
# Cluster analysis of European surface ozone observations for evaluation of MACC reanalysis data

Olga Lyapina[1], Martin G. Schultz[1], and Andreas Hense[2]

[1]Forschungszentrum Jülich, Institute for Energy and Climate Research: Troposphere (IEK-8), Jülich, 52425, Germany
[2]Meteorological Institute, Bonn University, Bonn, 53121, Germany

*Correspondence to*: Olga Lyapina (o.lyapina@fz-juelich.de) and Martin G. Schultz (m.schultz@fz-juelich.de)

**Abstract.** The high density of European surface ozone monitoring sites provides unique opportunities for the investigation of regional ozone representativeness and for the evaluation of chemistry climate models. The regional representativeness of European ozone measurements is investigated through a cluster analysis (CA) of 4 years of three-hourly ozone data from 1492 European surface monitoring stations in the Airbase database; the time resolution corresponds to the output frequency of the model that is compared to the data in this study. K-means clustering is implemented for seasonal-diurnal variations (i) in absolute mixing ratio units, and (ii) normalized by the overall mean ozone mixing ratio at each site. Statistical tests suggest that each CA can distinguish between 4 and 5 different ozone pollution regimes. The individual clusters reveal differences in seasonal-diurnal cycles, showing typical patterns of the ozone behavior for more polluted stations or more rural background. The robustness of the clustering was tested with a series of k-means runs decreasing randomly the size of the initial data set or lengths of the timeseries. Except for the Po Valley, the clustering does not provide a regional differentiation, as the member stations within each cluster are generally distributed all over Europe. The typical seasonal, diurnal, and weekly cycles of each cluster are compared to the output of the multi-year global reanalysis produced within the Monitoring of Atmospheric Composition and Climate (MACC) project. While the MACC reanalysis generally captures the shape of the diurnal cycles and the diurnal amplitudes it is not able to reproduce the seasonal cycles very well and it exhibits a high bias up to 12 nmol/mol. The bias decreases from more polluted clusters to cleaner ones. Also, the seasonal and weekly cycles and frequency distributions of ozone mixing ratios are better described for clusters with relatively clean signatures. Due to relative sparsity of CO and $NO_x$ measurements these were not included in the cluster analysis. However, simulated CO and $NO_x$ mixing ratios are consistent with the general classification into more polluted and more background sites. Mean CO mixing ratios are ≈ 140-145 nmol/mol (CL1 – CL3) and ≈ 130-135 nmol/mol (CL4 and CL5), and $NO_x$ mixing ratios are ≈ 4-6 nmol/mol and ≈ 2-3 nmol/mol, respectively. These results confirm that relatively coarse scale global models are more suitable for simulation of regional background concentrations, which are less variable in space and time. We conclude that cluster analysis of surface ozone observations provides a powerful and robust way to stratify sets of stations being thus more suitable for model evaluation.



# 1. Introduction

Tropospheric ozone is a strong oxidant affecting people's health and reducing yields of agricultural plants. Furthermore, it is responsible for a significant fraction of global warming (IPCC, 2013). Ozone is photo-chemically produced in the troposphere in a chain of chemical reactions from precursors which concentrations are strongly influenced by anthropogenic

activities. Maximum ozone concentrations are therefore often found in or near large urban agglomerations during summer (National Research Council, 1991) giving rise to the summer smog episodes. Since the 1990s tropospheric ozone has been continuously monitored at many ground sites across Europe. Numerical models of atmospheric transport and chemistry (CTMs) have become indispensable tools for the interpretation of measurement data, the analysis of sensitivities towards, for example, emission changes, and the evaluation of potential future air quality changes in the context of climate change.

Since 2005, a major European effort is under way to establish an operational system for monitoring and predicting global and European air quality with the help of data assimilation and numerical models (Hollingsworth et al., 2008). This Copernicus Atmosphere Monitoring Service (http://www.copernicus-atmosphere.eu/) has been developed in a series of projects funded by the European Commission under the acronym of Monitoring Atmospheric Composition and Climate (MACC). One of the products from MACC is a global reanalysis of atmospheric chemical composition covering the period

2003-2012 (Inness et al., 2013).

The quality of all model based estimates of atmospheric composition and its changes has to be assessed by in-depth model evaluations against observations. Currently model evaluation is often performed either on individual observations, or on the average of the set of measurements, selected from specific geographical regions. This is done for evaluation of global (Stevenson et al., 2006; Fiore et al., 2009; J.-F. Lamarque et al., 2012; Katragkou et al., 2015) as well as regional (van Loon

et al., 2007; Coman et al., 2012; Solazzo et al., 2012; Mailler et al., 2013) models or their ensemble. This approach is problematic because there is no guarantee that the regional average of selected stations gives a representative picture of the ozone distribution in that region. Furthermore there is large variability of ozone regimes even on small special scales and models will not be able to capture this variability unless they are run on very fine resolution. Therefore rather than aggregate data geographically we propose here to evaluate models based on groups of stations which share common characteristics

with respect to their ozone seasonal and diurnal cycles.

In the Airbase database (http://acm.eionet.europa.eu/databases/airbase/) more than 4000 stations from 39 European countries are classified based on the evaluation of the population distribution and emission sources in the proximity of the station. This scheme was defined  in the Council Decision 97/101/EC (EC Decision, 1997), which was revised and amended by Commission Decision 2001/752/EC (EC Decision, 2001), and finally modified by 2011/850/EU (EC Decision, 2011) as well

as described in Mol et al. (2008).

Analysis of the population distribution distinguishes the station type between urban, suburban or rural, while the assessment of emission sources in the surrounding area divides sites into traffic, industrial or background. Such categorization has the disadvantage of being based on subjective assessments by the different station maintainers or regional agencies. Moreover



the station information may become outdated: for example due to newly built industries, residential areas, roads or changes to forest areas. Such changes would transform stations from "background" to "urban", which would impede objective ozone analysis. Thus, a static category label as given in Airbase may not provide an objective and reproducible classification for use in further statistical analyses. Instead, we suggest applying cluster analysis (CA) to the measurement data as a data

driven classification. The main goal of this study is to identify typical European air quality ozone regimes, determine their indicative patterns with respect to the temporal behavior of ozone mixing ratios, to assess how well the classification works, and to apply the categorization to the evaluation of a global chemistry transport model. Analysis of groups separation was presented in Lyapina (2014) and won't be discussed here.

The output from the MACC reanalysis was sampled at all station locations, and the results were grouped into the same

clusters as the measurement data. Through comparison of the mean seasonal, weekly, and diurnal cycles and analysis of the variability of clusters, we can identify how well the MACC reanalysis can reproduce the ozone mixing ratios and seasonal-diurnal features of each regime, and as a consequence which regime is most consistent with the model results, and thus representative for the scale of the model.

The paper is structured as follows: section 2 describes the process of data filtering from the full Airbase database. The

extraction conditions for the MACC model data are given as well as further steps of the preparation of both data sets. Section 3 provides details about the applied k-means algorithm and Earth Mover's distance method. In section 4 the results of the two CAs are presented and compared to the MACC model data. Section 5 discusses the robustness of the cluster analyses and their application for the evaluation of models. Section 6 contains the conclusions.

## 2. Data

### 2.1. Airbase

Airbase provides hourly integrated ground-based ozone data records, measured by UV photometric analyzers. Geographically, the station network covers all countries from the European Union and the European Environment Agency (EEA) member countries (http://acm.eionet.europa.eu/databases/airbase/), albeit with varying density. Station altitudes vary from 0 to about 3100 m above sea level. In this study, Airbase version 6 data from 2007 to 2010 were used. Atmospheric

ozone content was recorded as ozone density in $\mu g \cdot m^{-3}$ units. For the analysis presented here these were converted to number densities (nmol/mol or ppb) using the density of dry air at $T_0 = 20$ °C and pressure $P_0 = 101325$ Pa. This $T_0$ corresponds to a conversion factor of 2 (i.e. 0.5 nmol/mol correspond to 1 $\mu g \cdot m^{-3}$ of ozone). $T_0 = 20$ °C and $P_0 = 101325$ Pa correspond to the standard settings of commercial ozone analyzers, which automatically convert measurements at actual temperature and pressure to these standard conditions.

Several datasets in Airbase contain incomplete data and some ozone records appear unreliable. Therefore a 4-step filtering procedure was applied to each dataset in order to identify suitable timeseries and to remove individual outliers which could corrupt the timeseries statistics. First, all data less than zero were eliminated, because they represent non-physical values.



Next, data above either 2.83 times the value of the 95%-quantile of the data or twice the value of the 99%-quantile were eliminated. For a Gaussian distributed random variable both values should be approximately identical. Even though the ozone probability density functions are generally not Gaussian (see Figure 9), this test can be used to define a reasonable upper limit value, because deviations from the normal distribution are mainly at the lowest percentile range of data. In a third

step those data points were removed which show erratic behavior near a missing value. The rationale behind this test is that a visual inspection of measurement timeseries sometimes indicates that data reporting stopped too late or resumed too early after a calibration procedure, an instrument maintenance or malfunction. On each side of the missing value, the five nearest measurements are tested if they lie in the range of the surrounding values or exhibit abnormal variability. Finally, another outlier test (multi-step low-pass filter) was performed using the 240 data points moving average in the first pass, which

removed data points exceeding 8 times the standard deviation within the moving sample. In the next two passes with a varying width between 10 and 72 points, thresholds of 8 and 6 standard deviations are applied.

The data filtering was tested extensively on many different ozone timeseries and found to reliably detect obvious errors while removing only very few valid data points. In order to retain a timeseries in the analysis it had to fulfill the following data capture criterion: in every year, at least 9 out of 12 months had to contain at least 2/3 of the theoretical maximum hourly

values. After application of this criterion, the original Airbase data set of more than 4000 stations was reduced to 1525 stations (see supplementary tables S1 and S2). Their timeseries were then visually inspected for sudden changes in the baseline (this phenomenon is not captured by the automated data quality filter; see also Solberg et al. (2009)). We adopted a conservative approach and flagged only those stations, where baseline shifts of 5 nmol/mol or greater occurred. The 33 stations which were filtered out at this step are presented in Table S2. Finally 1492 sites were used in the CA and model

evaluation (Table S1).

As input for the CA multi-annual monthly mean diurnal variations averaged over the four year period 2007 - 2010 of the individual ozone timeseries were used. Seasonal - diurnal ozone variations appear as typical cycles and represent the concentrations resulting from many factors influencing the particular stations. We used 3-hourly resolution rather than the original hourly resolution in order to match the frequency of the MACC model output (see section 2.2). Thus each station is

represented by a vector of dimension 96 (12 months times 8 time steps per day). The time averaged data at all stations were arranged as a data matrix of dimension 1492 by 96.

Two different input matrices for the CA were constructed leading to two different types of CA runs (1[st] CA and 2[nd] CA from here on). At first, seasonal - diurnal ozone variations in absolute values are used as a set of properties. Second, we used normalized seasonal - diurnal ozone variations in order to avoid the influence of actual ozone concentrations on the results.

Each normalized variation had zero mean and unit standard deviation. This second CA produces different clusters than the first step, but allocates stations to clusters being based on seasonal and diurnal variations themselves regardless of absolute concentrations. Since the data generally exhibit no trend during the 2007-2010 period and interannual variability is much smaller than the diurnal or seasonal variability, we did not detrend the data prior to the CA.





## 2.2. MACC

The model data were taken from the MACC reanalysis (Inness et al., 2013). The reanalysis invoked data assimilation of meteorological variables, and trace gas columns of $O_3$, CO, NO and $NO_2$ as well as ozone profile information from various satellite instruments. The model system was the European Centre for Medium Range Weather Forecasts (ECMWF)

Integrated Forecasting System which was coupled to the Model for Ozone and Related Tracers (MOZART) (Flemming et al., 2009; Stein et al., 2012). The model grid resolution was about 80 by 80 $km^2$, and 60 hybrid sigma-pressure levels covering the atmosphere from the surface to about 60 km altitude. Output was stored every 3 hours. We extracted gridded timeseries for the years 2007-2010. The model data at the 1492 Airbase stations used in the cluster analysis were obtained by a horizontal as well as vertical bi-linear interpolation to the locations and heights of the 1492 Airbase stations from the eight

nearest neighboring grid points. Similar to $O_3$, also CO, and $NO_x$ were extracted and provided as mole fractions.

For comparison with the CA results the model output was arranged in the same way as the Airbase observations (order of stations, and the set of properties). Then rows representing the reanalyzed trace gas concentrations at the observing stations were reordered according to the cluster membership of each station in the observation. In case of normalized set of properties the MACC data matrix was also normalized similarly to Airbase data and grouped according to the corresponding clustering

results of the 2[nd] CA.

## 3. Method

### 3.1. Cluster Analysis

Cluster Analysis (CA) is a data driven technique for classifying objects into groups whereby each object is described through a set of input parameters (properties or variables) which are used as criteria for grouping. Clusters are formed such that the

20 intra cluster similarity between objects inside a cluster and the inter cluster dissimilarity between objects of different clusters are jointly maximized. Initially the concept of "Cluster Analysis" was suggested by Tryon in 1939. Since then it has found applications in statistical processing of large data sets in biology, medicine, computer science, meteorology and atmospheric sciences (Zhang et al., 2007; Lee and Feldstein, 2013; Camargo et al., 2007; Christiansen, 2007; Beaver and Palazoglu, 2006; Dorling and Davies, 1995; Marzban and Sandgathe, 2006) as well as in other fields.

Several cluster algorithms have been developed and different choices can be made for the computation of distances between objects or groups of objects. The most commonly used types of clustering are hierarchical and partitional (aka. centroid-based clustering, or k-means clustering). Hierarchical clustering progressively splits the data set into more and finer clusters, whereas partitional clustering groups the data into a pre-determined number of clusters. Clusters are non-overlapping groups, such that at the end of the computation each object will belong to exactly one cluster. In the present study we applied

partitional clustering, because it allows for estimating the robustness of results and is less sensitive to outlier values than hierarchical clustering. K-means uses Euclidean metric $Edist$ for the calculation of distances:



$$Edist(A,B) = \sqrt[2]{\sum_{m=1}^{M}(x_{mA} - x_{mB})^2}, \tag{1}$$

where $x_A$ and $x_B$ are two objects of the data set each having M properties (i.e. variables); A and B are two different stations. In our case an object is a station timeseries of monthly averaged diurnal variations of three hourly ozone concentrations such that the Euclidean distance is evaluated from M=96 dimensions and is identical to the root mean square error between the two objects. The 1$^{st}$ CA uses absolute mixing ratio values, while in the 2$^{nd}$ CA the mixing ratios at each station are normalized by the mean so, that each object had zero ozone mean and unit standard deviation. The k-means algorithm minimizes the average Euclidean distances between individual objects and the given number of cluster centroids. A centroid C is an artificial object that represents its cluster and is the arithmetic mean of all properties of cluster members:

$$c_i = \frac{1}{n_i}\sum_{j=1}^{n_i} x_{ij}, \tag{2}$$

where $n_i$ – number of objects in $i^{th}$ cluster, $c_i$ – centroid of the $i^{th}$ cluster, $x_{ij}$ – $j^{th}$ object of the $i^{th}$ cluster. Minimization is achieved iteratively in an analysis cycle of three steps. At the initial step of each k-means run, k centroids are defined randomly from the data array. The second step assigns each object to the closest centroid by sorting in ascending order the distances $Edist(A, Ci)$. By this an initial seed of clusters is formed. In the third step, each centroid C is recalculated, as the mean of the current cluster members. Steps 2 and 3 are then repeated until the centroid coordinates don't change anymore. The goodness of the clustering can be assessed with the sum of squared distances (SSD) between all objects and their corresponding centroids:

$$SSD = \sum_{i=1}^{k}\sum_{j=1}^{n_k} Edist(c_i, x_{ij})^2, \tag{3}$$

where k is the number of clusters, and $n_k$ the number of objects inside the $k^{th}$ cluster. K-means requires that the number of clusters k is known for initialization of the algorithm, so prior to the CA we applied a method to determine the optimum value of k. Due to the random initialization, repetition of a k-means run with the same number of clusters will generate a sample of different SSD values as a function of the number of allowed clusters. Figure 1 shows an "elbow" - curve (SSD versus number of clusters k), derived from 50 · 100 independent k-means runs of the 1$^{st}$ set of properties (96 absolute seasonal-diurnal variations) with varying number of k from 1 to 100. The idea is to find the largest number of k where the SSD from the independent runs are consistent with each other, i.e. the curves in Figure 1 ideally fall onto a single point. For the first CA the optimum number of clusters is obviously k = 5. The "elbow" curves for the 2$^{nd}$ CA (Figure 2) suggest the use only 4 clusters in the analysis of normalized values.

The "elbow" plots not only give the appropriate number of clusters to run k-means, but they also provide a preliminary answer on the question of stability of the CA run for the chosen k. For the presentation of results in sections 4 and 5 we picked the k-means run with the lowest SSD out of the 100 independent realizations shown in Figures 20 and 21, respectively for each kind of CA. Further details on the stability (i.e. reproducibility) of k-means runs are given in section 5.



## 3.2. Earth Mover's Distance

In order to quantitatively evaluate the model's ability to reproduce the observed frequency distributions in each cluster, we calculated the Earth Mover's Distance (EMD). Initially the EMD was suggested by Rubner et al. (1998). EMD provides an objective distance measure between two frequency distributions or estimates of probability density functions. It is a true distance measure in the sense that it is positive semidefinite and symmetric and fulfills the triangle inequality. Additionally it has the property of being (asymptotically) proper meaning that the smallest distance is only achieved when the two probability densities are identical. The formula for EMD according to Rabin et al. (2008) is:

$$D(f||g) = \frac{1}{n_b} \sum_{i=1}^{n_b} |F_X(x_i) - G_X(x_i)|, \tag{4}$$

where $n_b$ is the number of bins, $F_{X(xi)}$ and $G_{X(xi)}$ are two cumulative distribution functions of f and g, which themselves are the two corresponding estimated probability densities obtained from the normalization of the respective frequency distribution histograms over the $n_b$ bins.

## 4. Results and Discussion

## 4.1. Geographical distribution and cluster allocation of stations

### 4.1.1. First CA

The spatial distribution of the 1492 Airbase stations and the respective cluster number of their classification obtained after the 1st CA are shown in Figure 3. Evidently, the five clusters do not simply represent different regions in Europe, although the members of cluster 1 (CL1) and cluster 2 (CL2) are concentrated in the Benelux and Ruhr region and in the Po Valley region, respectively. CL1 extends from Slovenia to Great Britain through the Netherlands, but also includes stations from France, Italy, Spain and Eastern Europe. Besides the Northern Italian stations CL2 also contains a few stations in the Alpine region, in the North-Western Balkans and in Spain. The third cluster (CL3) is much larger in its spatial extension and contains stations from almost all over Europe, including Scandinavia. The fourth cluster (CL4) spreads all over Europe with increased density along the Mediterranean coast and in the mountainous areas to the North and East of the Alps, the Bohemian Massif, and the Carpathian Mountains. Finally, the smallest cluster (CL5) largely overlaps with the mountainous regions of the Alps, the Pyrenees, Spain, and the Carpathians.

Table 2 presents a qualitative interpretation of the five clusters and shows the distribution of station altitudes for each cluster. The cluster descriptions were derived based on the geographical and altitude distribution together with a contingency analysis of the station type and station type of area attributes in the Airbase metadata. A contingency table with Airbase station attributes is provided in Table 1 (a,b). According to the Airbase classification (see "Introduction") stations are marked as either "urban", "suburban" or "rural" depending on the area type and as "traffic", "industrial" or "background" according to the station type. Each row in Table 1 corresponds to one of the Airbase clusters and shows the number of



stations related to each of 9 Airbase classification pairs. Most of the stations that we retained in our data filtering procedure (section "Data") are background stations, which could indicate that there are no local pollution sources in their vicinity. Measured concentrations should ideally be representative for a larger area (and hence suitable for the evaluation of numerical models), except when local effects from orography, land use or land-sea contrast confound the analysis. There is a relatively

even split between rural, suburban, and urban background stations. Industrial and traffic stations constitute about 10-15% each and are concentrated in the suburban and urban environments, respectively.

### 4.1.2. Second CA

Table 3 presents the same information as Table 2 but for the 2nd CA. There is some overlap between the cluster definitions of the 1st and 2nd CA. The 1st cluster of the 2nd CA corresponds to the 2nd cluster of the 1st CA, with the exception that it does

not contain stations from the Alpine region (Figure 4). The 2nd cluster is much larger and spreads over the Benelux and Ruhr regions in the Center of Europe, covering partly France, Switzerland and Eastern Europe, so it is partially overlapping with the 1st cluster from the 1st CA.

The 3rd cluster extends all over Europe and has several stations in Scandinavia. This cluster contains the largest number of stations. The 4th cluster includes high-mountain stations from the Alpine region and the Pyrenees, from the mountainous

areas to the North and East of the Alps, the Bohemian Massif, and the Eastern part of the Carpathian Mountains. Moreover it includes low- altitude stations from Spain, France, Great Britain, Scandinavia and the Mediterranean coast. Geographically it is a mix of stations from nearly all clusters of the 1st CA. The contingency tables with Airbase metadata (Table 1) and the geographical representation lead to the conclusion that the clusters from different CAs have some common features. For example, the 1st "Po Valley" cluster of the 2nd CA, which is mostly concentrated in the North of Italy, same as the 2nd cluster

of the 1st CA. The second cluster of the 2nd CA has the majority of stations, which were assigned to the first cluster in the 1st CA, and moreover captures also stations of the 2nd and 3rd clusters of the 1st CA. However, it appears as more elevated agglomeration. The 3rd cluster shares 326 stations out of more than 500 with the 3rd cluster of the 1st CA, resembling it also geographically and in altitude. It is the largest cluster in both CAs. The 4th cluster of the 2nd CA is containing both high and low altitude stations. It includes completely the 5th cluster and has some stations from 4th and 3rd clusters of the 1st CA.

Therefore on average the 4th cluster is semi-elevated with the mean altitude 433 m for the 2nd CA.

### 4.2. Comparison of Airbase clusters with MACC model results

### 4.2.1. Ozone means and consistency with ozone precursor concentrations

Figure 5 presents a comparison of the 5-25-50-75-95%- percentiles distributions from the 3-hourly Airbase and MACC initial data sets for the period 2007 – 2010 (i.e. length of each data set = 1492 stations · 4 years · 365 days · 8 values per day).

The mean and median volume mixing ratios averaged over the entire set of 1492 stations are 25 nmol/mol and 24 nmol/mol for Airbase, and 34 nmol/mol and 33 nmol/mol for MACC, respectively. Thus the 50% percentile and the mean of the model





data are both show a positive bias by 9 nmol/mol.

A more detailed pattern emerges when analyzing the station mean values using box and whisker plots separately for the five individual clusters of the 1st CA (Figure 6). With the exceptions of CL2 and CL3, which show quite similar distributions, the distributions of the observed (Airbase) values are rather distinct for each cluster and increase from CL1 to CL5. In

comparison, the MACC distributions are generally broader and exhibit a high bias of 5-12 nmol/mol except for CL5. MACC distributions also show increasing values from CL3 to CL5, but only little difference among clusters 1 to 3. Obviously, the model does not capture the differences among the somewhat more polluted sites very well. This is consistent with the distributions of simulated CO and NOx concentrations (there are too few observations available to make a meaningful comparison) shown in Figure 7. While the MACC model results show a clear separation between clusters 1-3 on the one

hand and clusters 4-5 on the other hand, they do not distinguish among CL1, 2, and 3. These results are not surprising given that ozone concentrations in CL1-CL3 are more likely influenced by local, small-scale pollution sources, which the model cannot simulate correctly with its grid point distance of approximately 80 km. It is however reassuring to see that the simulated mean values of ozone precursors are larger in those clusters that have been labeled more polluted according to the Airbase characterization tags.

Figure 8 shows the distributions of mean ozone mixing ratios in the clusters of the 2nd CA. The MACC distributions of mean values are again broader than the observations and the model overestimates all clusters with the highest bias of 14 nmol/mol for CL1 and the lowest 4 nmol/mol for CL4. The distribution of observed ozone means of CL4 is broader, than it is in the 1st CA. This can be explained by the mix of stations of various altitudes. For other clusters, the distributions are relatively narrow, but still nearly twice as broad as those of the 1st CA, except for CL1 (Figure 6). MACC model distributions of CO

and NOx concentrations for the clusters of the 2nd CA (not shown) are reflecting higher pollution levels in the first 2 clusters and moderate pollution conditions for CL3. CL4 is relatively clean and shows the lowest CO and NOx concentrations.

### 4.2.2. Frequency distributions of ozone in clusters

Above the comparison of ozone concentrations among the clusters and between the observations and the simulations was based upon quantiles characterizing the cumulative probability distribution. Another way are estimated probability density

functions or normalized frequency distributions computed by binning all available 3 hourly observations from both the Airbase and MACC data. Those frequency distributions are presented in Figure 9 for each cluster of the 1st CA and distinguished between summer and winter.

In the Airbase data the three clusters with more urban characteristics (CL1, CL2 and CL3) contain a significant number of values of very low concentrations, which are caused by ozone titration during the winter time in the presence of large

amounts of NOx from traffic and industries. Airbase winter distributions reveal peaks at low ozone mixing ratios with frequencies decreasing from CL1 to CL4, though the last is showing only few incidents of ozone titration. For clusters CL1, CL3 and CL4 the MACC model is able to capture some of this titration, but not for CL2 (Po Valley). No ozone titration occurs in CL5, neither in the observational data nor in the model results.





MACC exhibits quite a good fit to CL4 and CL5 winter ozone concentrations and in general shows a greater similarity with the frequency distributions of the observations in winter compared to summer. During summer the measured ozone data are almost normally distributed (except for CL1), which is not seen for the MACC summer values. The model summer curves exhibit a high bias and contain two maxima for CL2 and CL4 (Figure 9).

In order to quantitatively evaluate the model's ability to reproduce the observed frequency distributions in each cluster, we calculated the Earth Mover's Distance (EMD, described in section 3) (Table 4). As expected from Figure 9, the largest EMD is found for CL1 and CL2 in summer, while the model shows greater skill in capturing the frequency distributions of CL4 and CL5 and to a lesser extent also CL3 (Table 4). This is again consistent with the previous characterizations of CL3 as "background, moderately polluted" and of CL4 and CL5 as (mostly rural) background stations (Table 2). From CL1 to CL5

the EMD values for summer are decreasing, thus model prediction of observations improves in that order. We note that in the same order the level of pollution of clusters is decreasing while mean ozone concentrations are increasing. The winter EMD values are smaller than summer ones, and show no dependence from CL1 to CL5. In general the model describes winter ozone relatively well with the only exception of CL2, where MACC fails to predict the very low concentrations during titration events (Table 4, Figure 9).

Frequency distributions of the 3-hourly surface ozone values of Airbase and MACC for each cluster of the 2nd CA are presented in Figure 10. As anticipated from the previous discussion, clusters with urban signatures CL1 and CL2 are expected to show a peak at low ozone concentrations, related to their higher pollution level. Indeed, the peaks of Airbase probabilities of zero ozone concentrations are pronounced for both clusters in comparison to the moderately polluted CL3, for example, where "zero" ozone has twice less probability and the ozone maximum appears in the range 25-30 nmol/mol.

The shape of the relatively clean CL4 curve resembles a Gaussian distribution with maximum probability at ≈ 35 nmol/mol. EMD calculated for comparison of observations to modeled frequency distributions (Table 5) show the strongest disagreement for CL1, then follow CL2 and CL3 with quite similar values, and at the end is the smallest EMD value for CL4.

### 4.3. Analysis of seasonal, diurnal, and weekly variations

**4.3.1. First CA**

The mean seasonal amplitudes are defined as the difference between the highest and lowest 4-year average monthly mean ozone concentrations (Figure 11). This amplitudes estimated from the Airbase stations within the clusters of the 1st CA are generally between 18 and 24 nmol/mol (25%-ile to 75%-ile), with the exception of CL2 (Po Valley stations), where seasonal amplitudes range from about 26 to 37 nmol/mol (25%-ile to 75%-ile). The MACC model data show a similar pattern among

30 the clusters. However, the seasonal amplitude is often overestimated by 5-10 nmol/mol due to the overestimation of summer time ozone. The seasonal amplitude of CL2 stations is captured relatively well, although the mean values in CL2 exhibited the second highest bias (12 nmol/mol, Figure 6).



The seasonal cycles of the 1$^{st}$ CA cluster centroids are displayed in Figure 12. In the observations CL1 and CL3 run almost parallel and show a broad maximum extending from April to July for CL1 and a slight maximum in April for CL3. More prominent spring maxima are evident in CL4 and CL5, but CL5 also exhibits a second small peak in July. The only cluster with a single pronounced maximum in summer (July) is CL2. The spring maximum is typical for seasonal cycles of western

European sites and considered as Northern Hemispheric phenomenon (Monks, 2000). Indeed, a substantial subset of stations in CL3, CL4 and CL5 are situated along the western edge of the continent (see map, Figure 3). The decline of ozone mixing ratios from spring till autumn in CL3 and CL4 suggests that summer photochemical ozone formation plays only a minor role at these sites. On the other hand, the double peak of CL5 suggests a superposition of the "natural" spring maximum with the "anthropogenic" summertime photochemical ozone production. The stations in CL5 are more elevated, therefore they can be

influenced by ozone from the stratosphere-troposphere exchange, which is considered as a possible reason for the ozone spring maximum on high mountains (Elbern et al., 1997; Harris et al., 1998; Stohl et al., 2000; Monks, 2000; Zanis et al, 2003).

In contrast to the seasonal cycles of the Airbase cluster centroids, the cluster mean seasonal cycles of the MACC data all show a summer maximum of similar shape with peak in June. This suggests that either the summertime chemical ozone

formation is exaggerated in the model, or the largely transport-driven springtime maximum is underestimated. A potential influence from inconsistencies in the data assimilation (see Inness et al., 2013) is unlikely, but cannot be excluded.

The seasonal cycles in Figure 12 indicate that the MACC model performs better during winter than during the summer. This is particularly evident for clusters 3, 4, and 5, whereas a significant bias persists throughout the year for CL1 and CL2. In the Validation Report of the MACC reanalysis (2013) a comparison with GAW surface ozone data (Global Atmosphere Watch

program) shows that in most regions of the world ozone mixing ratios are generally underestimated during winter and overestimated during summer time. Inness et al. (2013) present an evaluation with EMEP data (European Monitoring and Evaluation Program, http://www.emep.int/), which is also consistent with this analysis. EMEP stations are almost exclusively characterized as background sites and are partly contained in the Airbase database as well.

Diurnal amplitudes were calculated from averaged diurnal cycles of each station as an absolute difference between daily

maximum and minimum, and then gathered into distributions for each cluster. Box and whisker plots of ozone average diurnal amplitudes (Figure 13) show a clear signature that appears to be correlated with the ozone precursor concentrations as simulated by the MACC model (see Figure 7). The largest diurnal amplitudes (mean 27 nmol/mol) are obtained for CL2 (Po Valley), followed by CL1 (mean 18 nmol/mol), CL3 (mean 18 nmol/mol) and CL4 (mean 17 nmol/mol). CL5 (relatively clean elevated) stations exhibit the lowest diurnal amplitude (mean 9 nmol/mol). This is consistent with earlier findings by

Flemming et al. (2005) and Chevalier et al. (2007), who show the smallest diurnal amplitudes for clean sites. The average diurnal amplitudes of the MACC model are generally consistent with the measurement data, except that the distributions are somewhat broader, and there is no big difference between the diurnal amplitudes in CL2 compared to CL1 and CL3. We note that the MACC model does not prescribe a diurnal cycle for ozone precursor emissions.

The diurnal cycles of the Airbase cluster centroids show rather similar patterns with peak values between noon and 15:00 h



for all clusters (Figure 14). CL2 shows the most pronounced maximum, while CL5 exhibits the flattest curve. Ignoring the overall bias the model diurnal cycles are similar to the observations except that ozone mixing ratios show a lesser decline from 00:00 h to 06:00 h in all clusters except for CL5. This could indicate underestimation of ozone dry deposition, possibly in conjunction with errors in the calculation of mixing in the nocturnal boundary layer. Underestimation of the diurnal amplitude in CL2 (Figure 13) is largely due to the model failure of capturing low ozone concentrations around 6 am (Figure 14).

Weekly amplitudes are shown in Figure 15. These were not used as initial parameters in the CA, but interestingly the classification of Airbase data shows a clear tendency of the weekly amplitudes decreasing from CL1 to CL5, even though there is considerable overlap between the various box-whisker plots. The weekly cycles of all cluster centroids show growth from Friday till Sunday, but no significant change during the week. This confirms our characterization of the clusters from more to less polluted, meaning that the less polluted sites are less influenced by local precursor emissions with distinct weekday cycles, notably traffic emissions (Beirle et al., 2003). As for the MACC model, the boundary conditions of its chemical equation system don't contain weekly variations of ozone precursor emissions, therefore simulated ozone has no significant weekly cycle.

Schipa et al. (2009) and Pollack et al. (2012) concluded that for polluted areas the higher ozone values during the weekend result from the fact that reduced NO emissions and relatively small changes in VOC emissions facilitate ozone production due to an increased VOC/NOx ratio. The median of weekly amplitudes in urban CL1 is 4 nmol/mol, which is consistent with Murphy et al. (2007). The MACC model results exhibit much smaller weekly amplitudes (generally less than 1 nmol/mol) with no apparent difference among clusters. It would be interesting to see how much of the weekly cycle can be produced by a global model if weekly variations of ozone precursor emissions were included, but this is beyond the scope of this study.

The large seasonal and diurnal amplitudes in the Airbase data of CL2 are consistent with the relatively large emissions and active photochemistry in the Po Valley region (Bigi et al., 2012). While ozone precursor concentrations at stations in CL1 may be as large as those in CL2 (based on emission inventories and the MACC simulation results for CO and NOx, see Figure 7), the mean ozone concentrations at these stations are lower. As can be seen from the frequency distributions in Figure 9, there are a lot more incidents with very low ozone concentrations at the stations in CL1, and these occur both in winter as well as in summer. In the Northern and Central part of Europe, where the majority of CL1 stations are located, the photochemistry is slow especially during winter, so that not much NO2 is converted back to NO and ozone via photolysis. CL2 also exhibits ozone titration, but in summer to a lesser extent than for CL1 (Figure 9). For CL2 ozone destruction by NO and dry deposition still occurs during night time but in contrast to CL1 this is compensated by elevated ozone concentrations from photochemical production during daytime. In addition, the seasonal cycle is more pronounced for CL2 than for CL1 (Figure 12). This may be explained by the basin type of the Po Valley region and by its partly sub-tropical climate with plenty of available UV light, which is favorable for summer diurnal photochemical ozone production.




### 4.3.2. Second CA

The mean seasonal amplitudes for clusters of the 2$^{nd}$ CA are presented in normalized units in Figure 16. MACC data was normalized in the same way as the Airbase data, and then grouped according to the clustering results. We notice narrowness of seasonal amplitudes distributions and the decrease of their average in order CL1 → CL2 → CL3 → CL4. MACC seasonal

amplitudes follow the same dependence, but in a more "smoothed" way, and they have broader distributions. The means of modeled amplitudes slightly overestimate average observed amplitudes for CL3 and CL4, are nearly equal for CL2 and underestimate CL1.

The seasonal cycles in normalized values of the cluster centroids from the 2$^{nd}$ CA are depicted in Figure 17. In contrast to the results from the 1$^{st}$ CA, the seasonal cycles of centroids show gradual change from the smoothest cycle of CL4 ("background

rural") with only April maximum to the most prominent cycle of CL1 ("background urban") with strong July maximum. CL2 presents an intermediate cycle with a broad maximum, and CL3, although it has a more pronounced amplitude than CL4, still preserves the same features with a dominant spring peak. While the annual amplitudes are generally well described, the model cannot distinguish different seasonal patterns, like spring maximum or July peak, but always presents broad symmetrical bell-shaped summer maxima. The model underestimates normalized seasonal cycles in the beginning of

the calendar year (except for CL1) and spring time as well as overestimates in autumn for CL1 and 2 and also in summer for CL3 and 4.

With respect to seasonality the best match between model and observations is found in CL3 and CL4. Some underestimation in spring and winter is evident for CL3, and though in summer time there is a good fit of diurnal cycles in daytime, the observations show more ozone titration during the night, which is not captured by the model. The least well predicted

centroid of CL1 has strong disagreements between model and observations. Box and whisker plots of average diurnal ozone amplitudes expressed in normalized values (Figure 18) are continuously decreasing in their mean from CL1 to CL4, likewise the distributions of seasonal amplitudes (Figure 16). For all clusters modeled ozone diurnal amplitudes distributions are broader and underestimating observed ones.

In general the model performs better for the description of diurnal cycles rather than seasonal. The diurnal cycles (Figure 19)

give also similar dependence on cluster number as seasonal cycles: the smoothest for CL4 and most pronounced for CL1. As expected from the 1$^{st}$ CA, all clusters exhibit diurnal minima at 6 am and maxima between midday and 3 pm, except for CL1, which maximizes in the late afternoon - after 3 pm, similarly to CL2 of the 1$^{st}$ CA. Modeled diurnal minima and maxima are in accordance with the observations, except for CL1, where MACC shows daily maxima in between 12 and 3 pm like for other modeled groups.

Clustering based on the normalized set of properties gives as a result a clear division of stations relevant to amplitudes of seasonal and diurnal cycles (Figures 16, 18). Further analysis (not presented here) of the 2$^{nd}$ CA clusters have shown that they are also distinguished by the short-term variability, expressed as difference between 95- and 5-percentiles of ozone mixing ratios (Lyapina, 2014). Both these amplitudes as well as variability decrease uniformly and gradually from CL1 to



CL4 in accordance with the level of pollution of these clusters. In contrast, there are no substantial differences of variability between clusters of the 1$^{st}$ CA (Lyapina, 2014). And as mentioned earlier, the dominant clustering criteria of the 1$^{st}$ CA are the average ozone concentrations (Figure 6), and only to lesser extent the seasonal-diurnal amplitudes.

## 5. Stability and robustness of the cluster analyses

As described in section 3.1 "Cluster analysis", repeated k-means runs do not necessarily lead to the same allocation of stations to clusters due to the random assignment of the initial centroids. As explained there, different initialization may lead to somewhat better or worse separation of clusters as expressed by the SSD values. Here we analyze the reproducibility of results from many independent k-means runs. We call this the stability of the CA. Another important aspect investigated here is the robustness of the analysis, i.e. the reproducibility of the station classification when random subsets of stations or are

excluded from the analysis or when the input data are shortened in time.

### 5.1. Stability of the CA

As mentioned in section 3 "Method", 100 independent k-means runs were carried out for each CA and from these runs the one with the smallest SSD was chosen for further analysis. These runs (one for the first and one for the second CA) will be referred to as reference runs.

The plot of the SSD values for each of these 100 runs of the 1$^{st}$ set of properties (Figure 20) reveals at least three "stable states" with 75 realizations out of 100 yielding smaller SSD values, a few cases with moderate SSD and about a quarter of realizations with much larger values. All of the 75 runs with smaller SSD generate a very similar classification of stations: 4 runs (green dots in Figure 20) with more than 99% identity to the reference run and 70 runs with more than 95% of stations are grouped into the same categories as in the reference case which is marked with a black diamond in Figure 20. The

stability decreases when the SSD values become larger, but in all of the runs at least 89% of the stations are always classified in the same way. Exemplary checks of how the stations are redistributed when the results differ indicate that we usually find CL3 stations from the reference run in CL1 and CL2, while some CL4 stations are moved to CL3. This indicates that the distinctions between these clusters may be less obvious if we base our analysis on mean concentrations as we did in this study.

Similar to Figure 20, Figure 21 shows the SSD values of the 100 k-means runs from the second CA. From the first look at Figure 21 we notice that the SSD curve of 100 k-means runs based on 2$^{nd}$ set of properties is less structured and exhibits no "stable states". However, the scale of SSD values is also very narrow here, and every run generates a classification which is at least 95% similar to the reference run of the second CA.

### 5.2. Robustness with respect to number of stations considered

Besides the 100 k-means runs with all 1492 stations we performed another 100 sets of 100 k-means runs each where we





randomly reduced the number of stations to 90, 80, 70, 60 and 50% of the initial data set, respectively. For each of these sets we selected the run with the minimum SSD and compared the classification results with our reference run. The robustness of the CA results was then obtained from contingency tables, where diagonal elements reveal the number of stations that are classified to the same cluster as in the reference run.

Table 6 summarizes the results of all of these tests by grouping the contingency results into three categories: better than 99% agreement, 95-99% agreement, and less than 95% agreement of cluster allocations (in this case there were no cases with less than 89% agreement for k-means runs of the 1$^{st}$ set of properties). Each row in Table 6 represents the results for one particular dataset size. As Table 6 shows, the CA classification is very robust (more than 95% agreement in 99 runs out of 100) even if only 60% of the stations remain in the dataset. Out of the 100 randomly selected subsets for each row, at least
25 yield a classification which is 99% consistent with the reference run. Only if we remove 50% of the stations from the input data, this similarity starts to decline. Note again that each count in Table 6 is already the minimum SSD run out of 100 for a given random sample. Had we performed only one realization of each subset, the CA would appear much less robust because of the stability issues discussed above.

Table 7 shows the robustness results of the second CA. Though the reproducibility of 2$^{nd}$ CA runs with the full data set is
higher (see Figure 21) than runs based on the 1$^{st}$ set of properties (Figure 20), the reduced data sets give the opposite results. Reduced till 70% data set delivers most of 2$^{nd}$ CA runs into the second category (95-99% of similarity), what happened only for the half-size reduced data set of the 1$^{st}$ CA runs. Nevertheless, in case of the 2$^{nd}$ set of properties no single run produces less than 91% agreement with the reference run, which is slightly better than for the 1$^{st}$ set of properties (89% of similarity). But as there are very few such runs (maximum 8 runs out of 100) in both CAs, we can conclude that the most of runs with
any reduction result in clustering with 95% and higher similarity to the reference runs.

### 5.3. Robustness with respect to the length of the timeseries

Obviously it is desirable to obtain a station classification which is independent of the precise time period that is chosen for the analysis. We therefore performed additional robustness tests of the two CAs by repeating the analysis for subsets of 3 years out of the total 4 years which we had available. Each CA was re-calculated in 4 sets of 100 realizations excluding all
data from 2007, 2008, 2009 and 2010, respectively. As before, from each set the run with minimum SSD was selected and compared to the reference runs. The similarities of the station classification were again taken from the diagonals of contingency tables and are given in Table 8. There are small differences depending on which year is removed from the analysis, and on average both CAs yield a classification which is 95% similar to the analysis of the complete dataset.

### 6. Conclusions

Starting from more than 4000 European Airbase surface stations monitoring ozone concentration for the period 2007 till 2010 finally 1492 were selected after filtering for incomplete timeseries and erroneous data. The classification of stations




based on k-means cluster analysis is broadly consistent with the Airbase intrinsic description of area types, which divides station types into background, industrial and traffic and station area types into urban, suburban and rural. The consistency between this Airbase characterization and our classification is mainly reflecting the pollution levels in the individual clusters. From the chosen parameters for the investigation of ozone representativeness, namely absolute as well as normalized

seasonal-diurnal variations provided as monthly averaged diurnal cycles with 3-hour time resolution, 5 and 4 clusters respectively yield the most stable clustering results. Most of these clusters spread across the entire European domain. This implies that differences in the local setting of stations (altitude, anthropogenic emissions) are more important than the geographic location for characterizing the seasonal-diurnal ozone cycles. Because of the strong spatial overlap between clusters the representativeness of different ozone air quality regimes is not related to the territory covered by the stations set

of any cluster. It indicates that comparison with a model based only on a geographical basis would not lead to an informative validation of model prediction of typical ozone regimes. Cluster analysis is a valid tool for obtaining clearer and more interpretable results for MACC validation.

In the first cluster analysis (1$^{st}$ CA) based on absolute seasonal-diurnal variations stable results are obtained with a classification into 5 clusters (CL1 – CL5). Differences in the seasonal cycles among the clusters reflect typical patterns of the

ozone behavior in traffic, urban, suburban, rural and elevated regions. The first 3 clusters represent more polluted regimes, while the other two exhibit characteristics of more rural and clean sites. This interpretation is supported by comparing simulated concentrations of the precursor CO and $NO_x$ from the MACC reanalysis and the frequency distributions of hourly ozone values in clusters.

The seasonal cycles of the 2$^{nd}$ CA show a gradual change from the smoothest cycle of CL4 with a maximum in April to the

most pronounced cycle of CL1 with a strong July maximum. CL2 presents intermediate conditions with a broad maximum, and CL3, although it has a more pronounced amplitude than CL4, still preserves the same features with a dominant spring peak. Diurnal cycles exhibit similar tendencies with a more pronounced cycle in CL1 and a flat one in CL4. In the 1$^{st}$ CA clusters are distinguished first of all by the mean ozone concentrations, and as a consequence, station altitudes play a major role. In contrast, using the same set of properties with normalized values (2$^{nd}$ CA) the seasonal and diurnal amplitudes

dominate the clustering.

The ozone variability (expressed as difference between 95- and 5-percentiles) was not included as an input parameter for any of the CAs. As an outcome there are no substantial differences of variability between clusters of the 1$^{st}$ CA. In contrast, for the CAs based on the normalized properties the variability reduces from CL1 to CL4 (Lyapina, 2014). This implies that the short-term variability of ozone concentrations at European stations is generally correlated with the seasonal and diurnal

amplitudes at these sites.

Comparison of the model with observations for individual clusters reveals MACC deficits and pros. First of all there are different overestimation biases for the 1$^{st}$ CA (from $\approx$ 5 to $\approx$ 15 nmol/mol), and secondly differences mainly in seasonal behavior rather than diurnal for both 1$^{st}$ and 2$^{nd}$ CAs. The biases are mostly driven by summertime ozone rather than wintertime, where ozone is generally well predicted (biases less than 5 nmol/mol on average). The biases decrease when





going from clusters indicative of higher pollution to cleaner ones. Also, the seasonal cycles are described better for clusters with relatively clean air signatures. The best fit between the MACC reanalysis and the observations is observed for CL5 of the 1st CA as well as for CL4 of the 2nd CA and is explained by the fact that these stations are influenced more by regional rather than local factors.

When applying the k-means clustering technique it is important to ensure that the results are stable and robust against spatial and temporal subsampling of the data array. We analyzed the reproducibility of the clustering results based on an extensive number of repetitions and found that in general, more than 95% of stations are mostly always grouped into the same category, even if the total number of stations is reduced to 60% of the total, or if one year is excluded from the analysis. However, this robustness is only obtained if one performs several k-means runs for each subset and selects the run with

minimum SSD for further analysis. We therefore conclude that k-means clustering presents a suitable analysis of ozone mixing ratio data if applied in the described manner.

The robustness and clarity of the cluster analysis might be further improved by adding observations of other compounds (ozone precursor concentrations) and/or meteorological variables. Unfortunately, such data are only available for very few of the Airbase measurement sites. Inclusion of such data might also allow separation into more clusters where one might begin

to see regional differences of the ozone behavior. As the robustness analysis indicates, our results should remain valid even if the analysis were to be repeated with longer timeseries or with an extended or reduced set of stations. It would be interesting to perform similar analyses in other world regions and to find out if the clusters obtained there are related to the broad classification into pollution regimes which we found for Europe.

**Acknowledgements**

We are grateful to the European Topic Centre on Air Pollution and Climate Change and Mitigation (ETC/ACM) on behalf of the European Environment Agency for managing Airbase, to the MACC-II project team for designing and operating the MACC forecasting system and running the reanalysis and to EU for funding MACC-II under grant no. 283576. We also thank S. Waychal for programming support, and O. Stein and S. Schröder for processing of the MACC reanalysis dataset. Besides, we are very thankful to the Jülich Supercomputing Center for letting us run model simulations.



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





Table 1. (a) Contingency table, showing the distribution of stations in clusters (rows) versus Airbase classification groups (columns). Abbreviations: Bac – background, Ind – industrial, Trf – traffic, Rur – rural, Sub – suburban, Urb – urban. 1$^{st}$ CA. Colors: no color if value < 10, light blue for 10 – 50, blue 50 – 100 and dark blue > 100.

| CL | BacRur | BacSub | BacUrb | IndRur | IndSub | IndUrb | TrfRur | TrfSub | TrfUrb | total |
|---|---|---|---|---|---|---|---|---|---|---|
| 1 | 30 | 78 | 134 | 3 | 22 | 11 | 6 | 13 | 85 | 382 |
| 2 | 22 | 45 | 64 | 2 | 6 | 3 | 1 | 3 | 9 | 155 |
| 3 | 117 | 147 | 184 | 12 | 20 | 11 | 1 | 4 | 28 | 524 |
| 4 | 135 | 53 | 50 | 16 | 22 | 10 | 0 | 3 | 15 | 304 |
| 5 | 103 | 12 | 1 | 5 | 3 | 1 | 0 | 0 | 2 | 127 |
| total | 407 | 335 | 433 | 38 | 73 | 36 | 8 | 23 | 139 | 1492 |
| | Bac | 1175 | | Ind | 147 | | Trf | 170 | | |
| | Rur | 453 | | Sub | 431 | | Urb | 608 | | |

(b) Same as above, but for the 2$^{nd}$ CA.

| CL | BacRur | BacSub | BacUrb | IndRur | IndSub | IndUrb | TrfRur | TrfSub | TrfUrb | total |
|---|---|---|---|---|---|---|---|---|---|---|
| 1 | 14 | 25 | 56 | 0 | 1 | 0 | 0 | 1 | 11 | 108 |
| 2 | 46 | 136 | 154 | 6 | 29 | 11 | 6 | 10 | 58 | 456 |
| 3 | 129 | 140 | 162 | 17 | 30 | 15 | 2 | 10 | 46 | 551 |
| 4 | 218 | 34 | 61 | 15 | 13 | 10 | 0 | 2 | 24 | 377 |
| total | 407 | 335 | 433 | 38 | 73 | 36 | 8 | 23 | 139 | 1492 |
| | Bac | 1175 | | Ind | 147 | | Trf | 170 | | |
| | Rur | 453 | | Sub | 431 | | Urb | 608 | | |





Table 2. Cluster statistics and description based on the Airbase classification, geographical location and altitude range of clusters. 1$^{st}$ CA.

| CL | cluster description | amount of stations | mean altitude, m (25..75 percentiles) |
|---|---|---|---|
| 1 | urban traffic | 382 | 177 (35..250) |
| 2 | urban/sub-urban, Po Valley | 155 | 243 (72..381) |
| 3 | urban/sub-urban | 524 | 203 (50..287) |
| 4 | rural/industrial/remote, middle-elevated | 304 | 288 (45..503) |
| 5 | rural background, elevated | 127 | 819 (370..1137) |



Table 3. Cluster statistics and description based on the Airbase classification, geographical location and altitude range of clusters. 2$^{nd}$ CA.

| CL | cluster description | amount of stations | mean altitude, m (25..75 percentiles) |
|---|---|---|---|
| 1 | Po Valley, urban, traffic | 108 | 200 (45..293) |
| 2 | urban/sub-urban, industrial, traffic | 456 | 250 (90..360) |
| 3 | moderately polluted (urb., sub., rur.), industrial, traffic | 551 | 190 (35..273) |
| 4 | rural, remote, coastal, background, middle -elevated, industrial | 377 | 433 (35..735) |



Table 4. EMD values for each cluster between Airbase and MACC data (2007-2010). $1^{st}$ CA.

| CL | summer | winter | all |
|----|--------|--------|-------|
| 1  | 0.181  | 0.068  | 0.126 |
| 2  | 0.146  | 0.112  | 0.134 |
| 3  | 0.139  | 0.028  | 0.083 |
| 4  | 0.110  | 0.021  | 0.064 |
| 5  | 0.092  | 0.025  | 0.041 |



Table 5. EMD values for each cluster between Airbase and MACC data. 2$^{nd}$ CA.

| CL | EMD (obs-mod) |
|----|---------------|
| 1  | 0.15          |
| 2  | 0.106         |
| 3  | 0.091         |
| 4  | 0.051         |



Table 6. Robustness analysis of the 1$^{st}$ CA. The table lists the number of k-means runs (out of 100) where stations are assigned to the same cluster as in reference run after reducing the dataset by randomly removing 10, 20, 30, 40, and 50% of stations, respectively. Categories show percent of similarity to the reference run, i.e. number of stations clustered to the same group as in reference run. Category 3 has the lowest similarity, but at least 89% of the stations were reproducibly assigned to the same clusters.

| data | | results | | |
|---|---|---|---|---|
| fraction of data,% | number of stations | >= 99% (cat.1) | 95 - 99% (cat.2) | < 95% (cat.3) |
| 90 | 1343 | 86 | 14 | 0 |
| 80 | 1194 | 46 | 53 | 1 |
| 70 | 1044 | 39 | 58 | 3 |
| 60 | 895 | 25 | 74 | 1 |
| 50 | 746 | 12 | 80 | 8 |





Table 7. As table 6, but for the 2$^{nd}$ CA. Here, no runs occurred where less than 91% of stations were assigned to the same cluster as in the reference run.

| data | | results | | |
|---|---|---|---|---|
| fraction of data,% | number of stations | >= 99% (cat.1) | 95 - 99% (cat.2) | < 95% (cat.3) |
| 90 | 1343 | 62 | 38 | 0 |
| 80 | 1194 | 62 | 38 | 0 |
| 70 | 1044 | 13 | 86 | 1 |
| 60 | 895 | 13 | 86 | 1 |
| 50 | 746 | 9 | 83 | 8 |



Table 8. Similarities (percentages of stations assigned to identical clusters) between reference CA runs and runs based on data sets with excluded years (see text).

| set of properties | missing year | | | | |
|---|---|---|---|---|---|
| | -2007 | -2008 | -2009 | -2010 | average |
| 1st | 94.8 | 95.1 | 95.0 | 94.9 | 95.0 |
| 2nd | 93.0 | 96.3 | 96.2 | 94.0 | 94.9 |





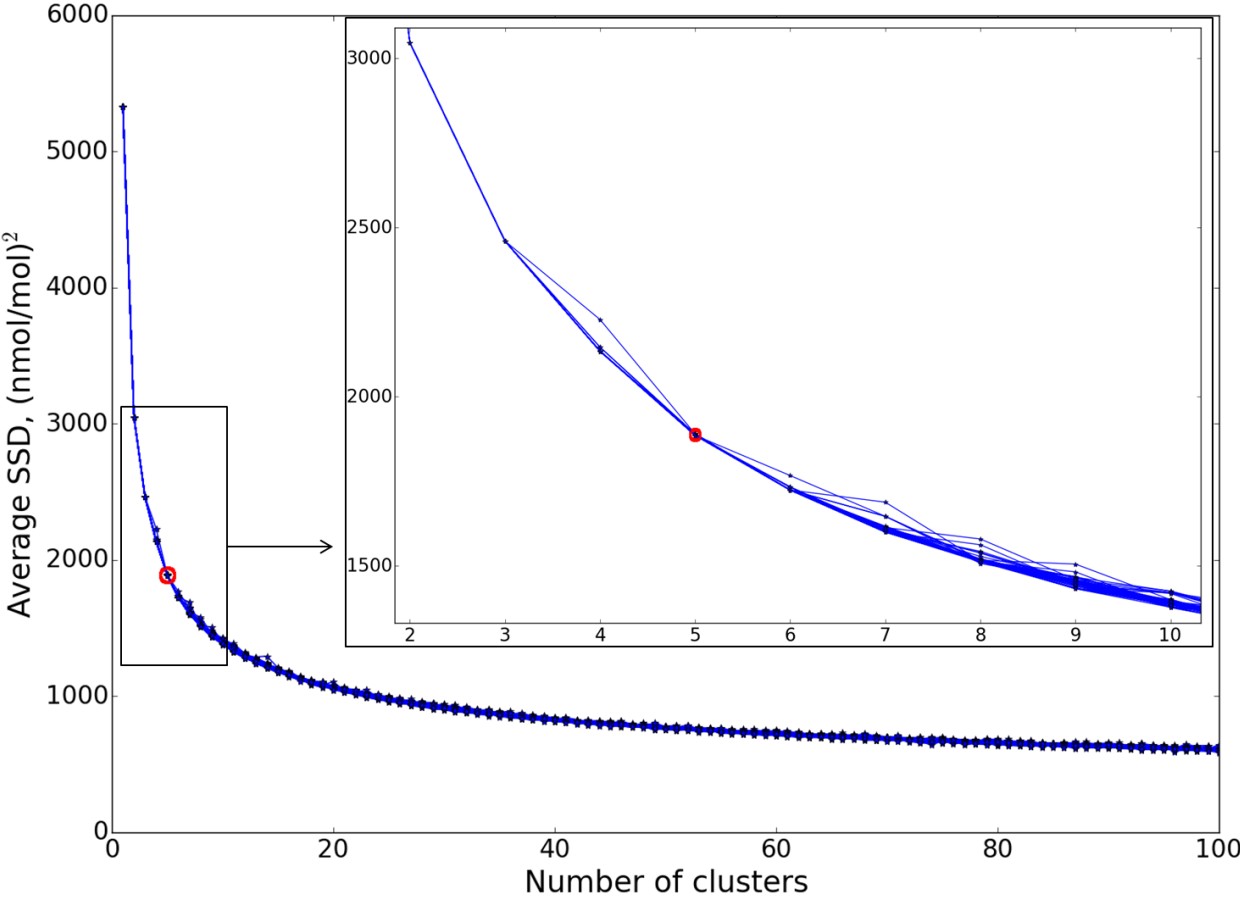

Figure 1. Averaged SSD ("elbow"- plot) of 50 · 100 independent k-means runs with varying number of clusters k from 1 to 100, based on the 1[st] set of properties.





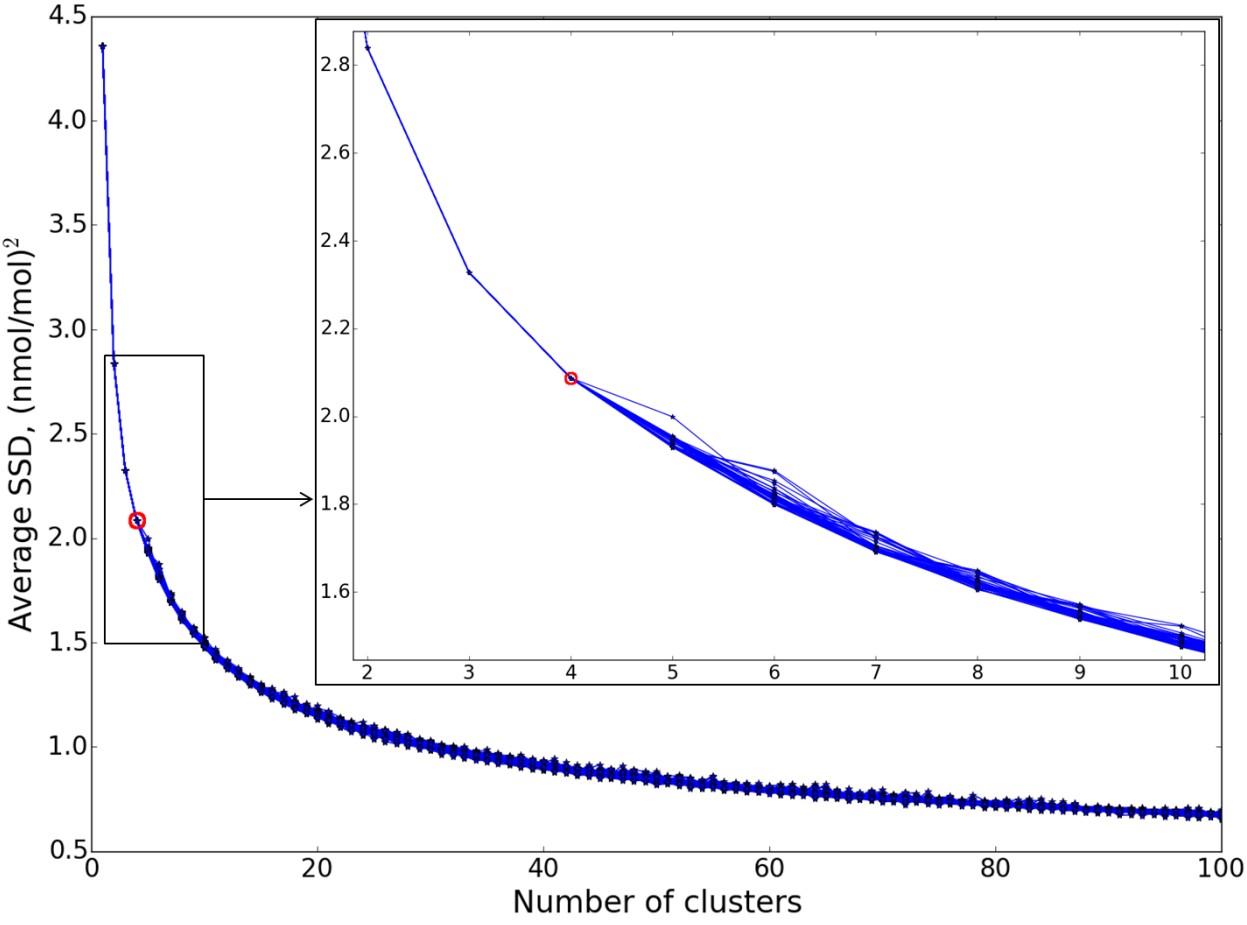

Figure 2. Averaged SSD ("elbow"- plot) of 50 · 100 independent k-means runs with varying number of clusters k from 1 to 100, based on the 2$^{nd}$ set of properties.



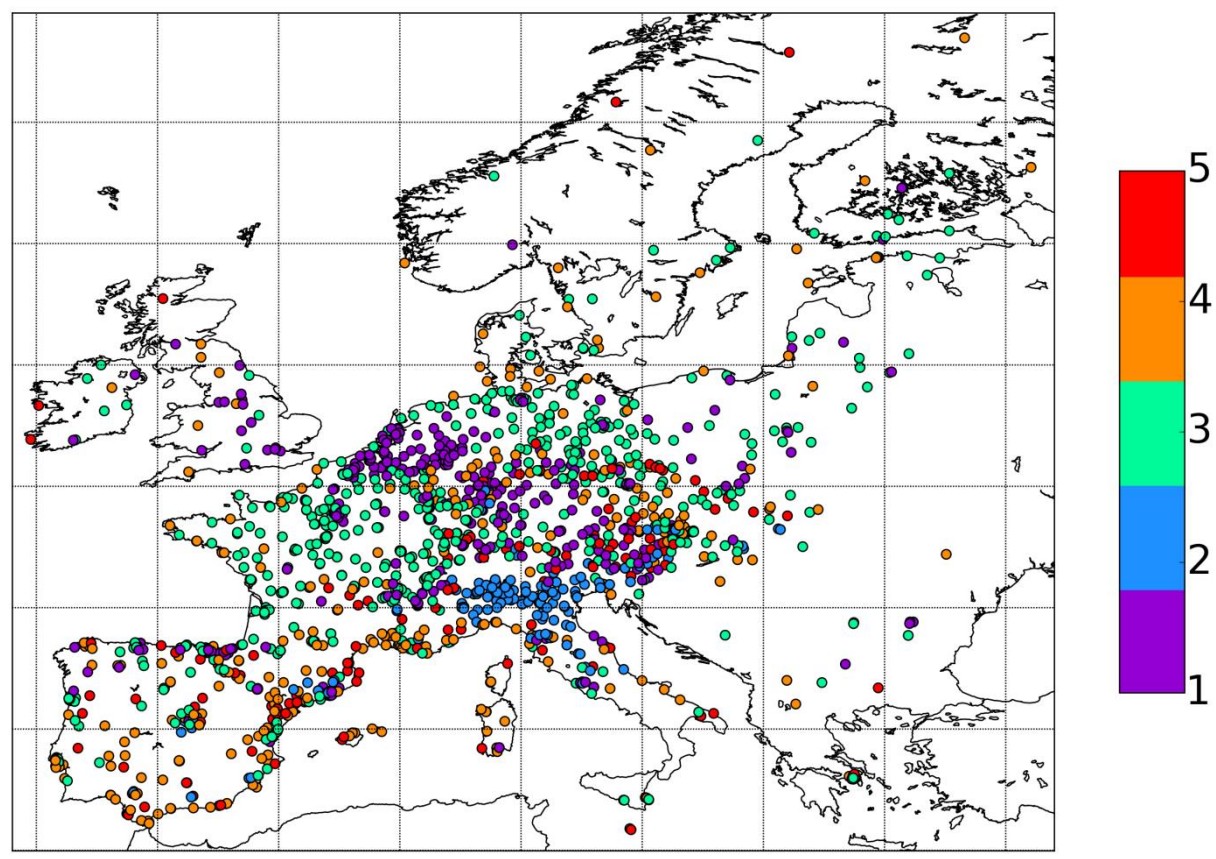

Figure 3. Map of 1492 Airbase stations clustered in 5 groups. 1st CA.





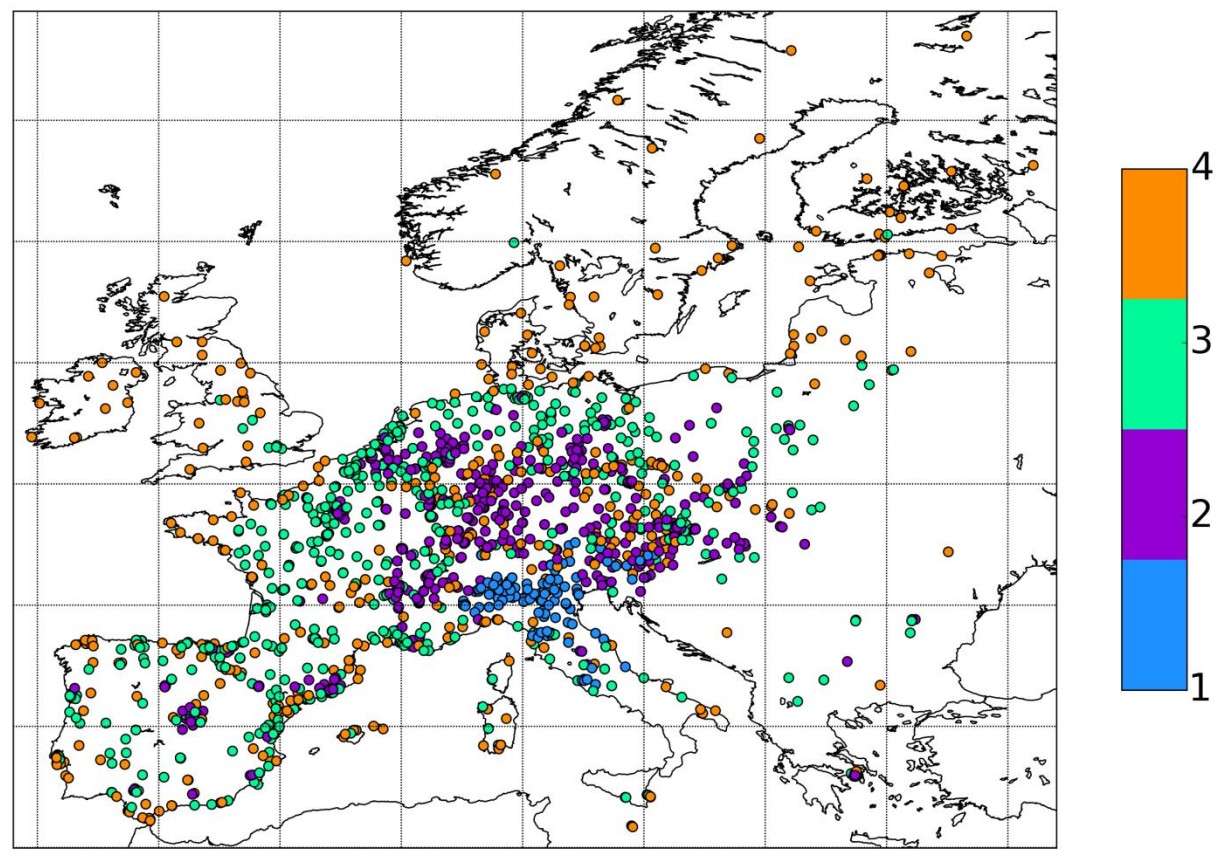

Figure 4. Map of 1492 Airbase stations clustered in 4 groups. 2$^{nd}$ CA.





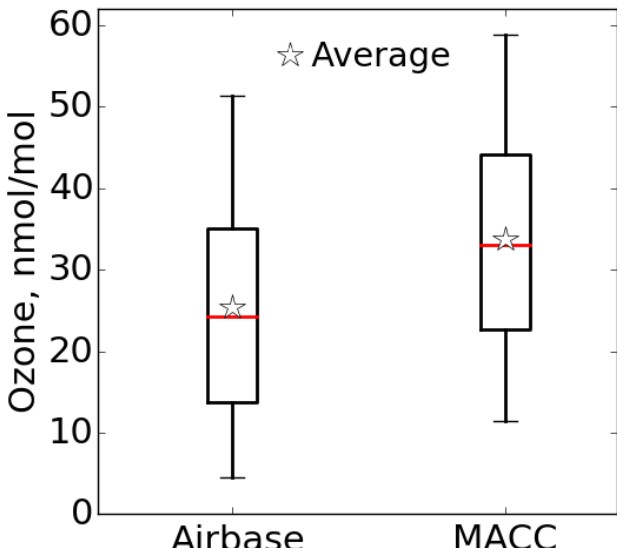

Figure 5. Percentiles (5-25-50-75-95) of 3-hourly ozone mixing ratios for 1492 stations, Airbase vs MACC.



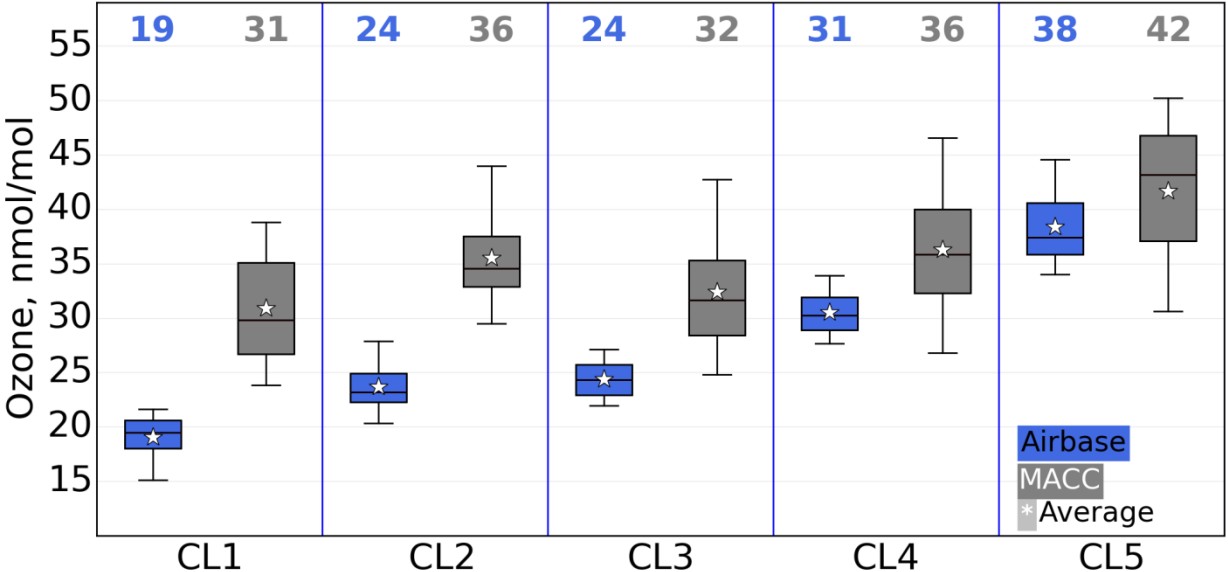

Figure 6. Percentiles (5-25-50-75-95) of ozone means in clusters, Airbase vs MACC. Upper values indicate the mean of each cluster. 1st CA.





(a)

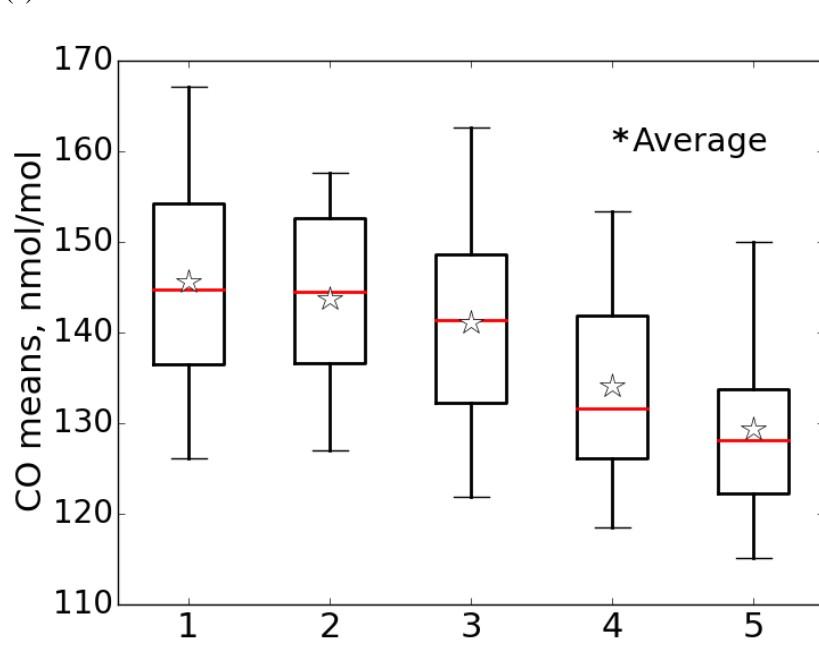

(b)

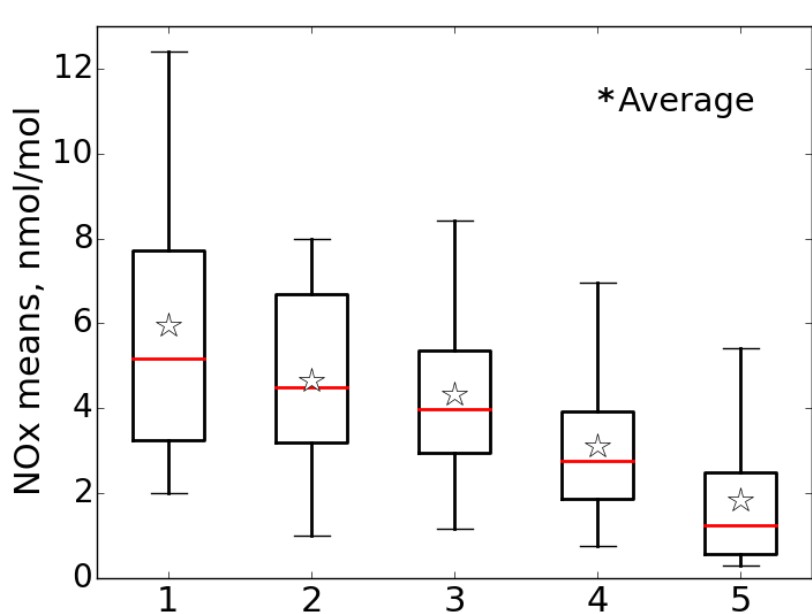

5    Figure 7. Percentiles (5-25-50-75-95) of modeled CO (a) and NOx (b) means in clusters. 1[st] CA.





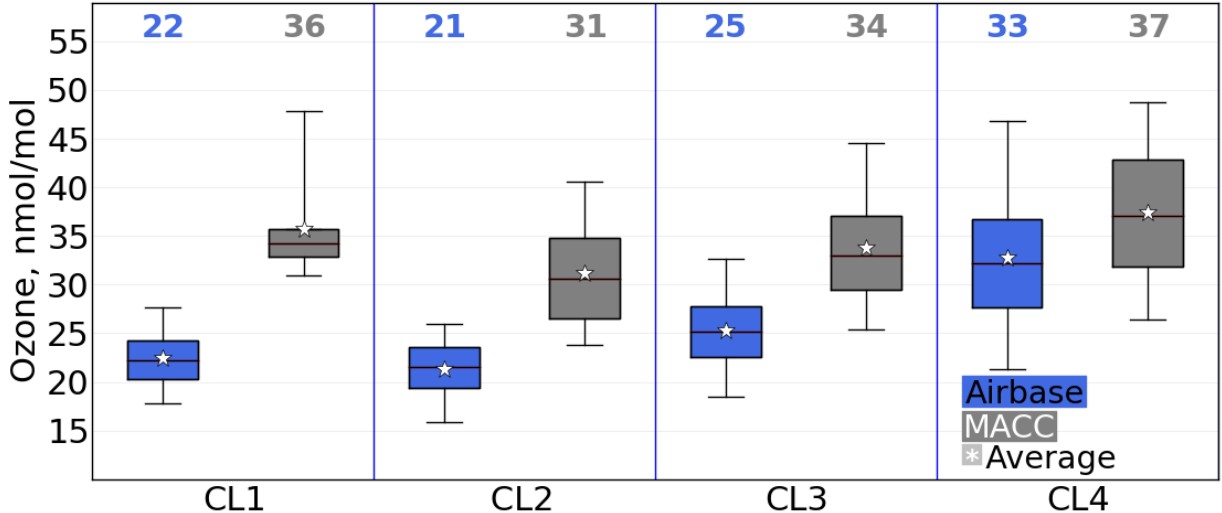

Figure 8. Percentiles (5-25-50-75-95) of ozone means in clusters, Airbase vs MACC. Upper values indicate the mean of each cluster. 2nd CA.





Figure 9. Normalized frequency distributions of 3-hourly ozone values in clusters (2007-2010), summer (left) and winter (right), Airbase vs MACC. 1$^{st}$ CA.





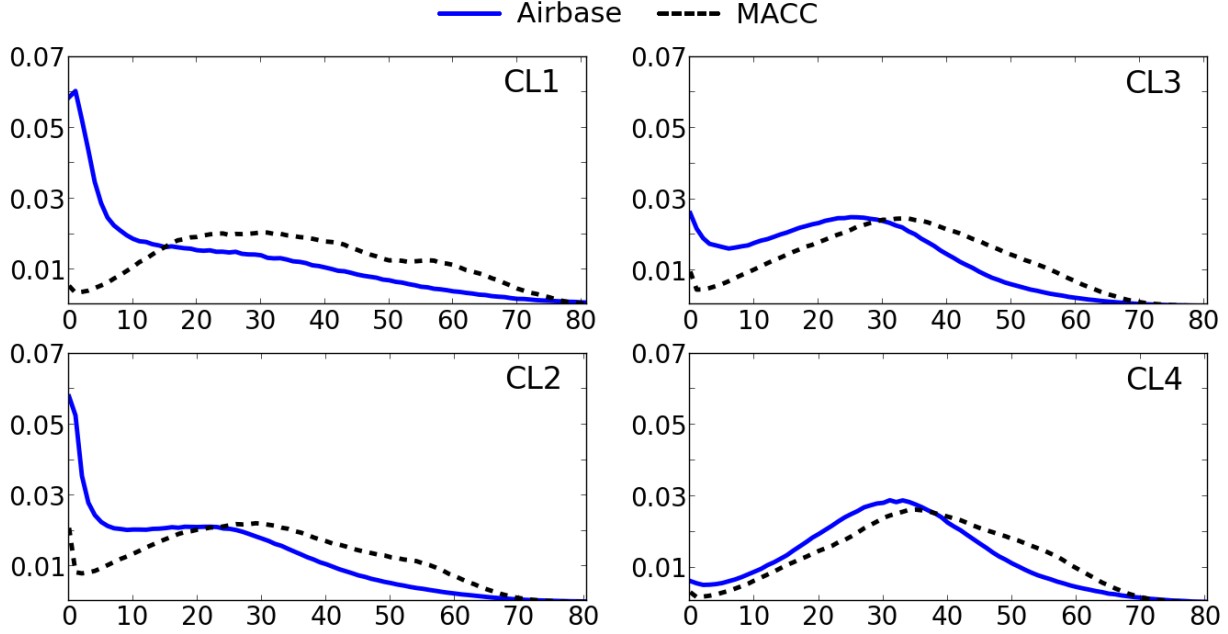

Figure 10. Normalized frequency distributions of 3-hourly ozone values in clusters (2007-2010), Airbase vs MACC. 2$^{nd}$ CA.





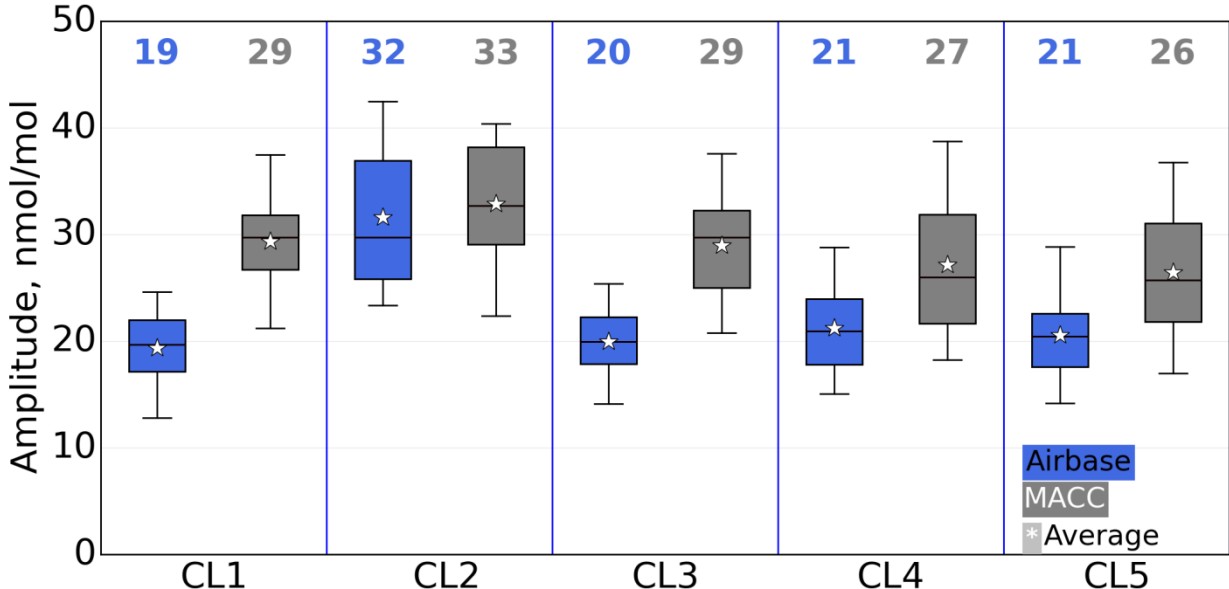

Figure 11. Percentiles (5-25-50-75-95) of ozone seasonal amplitudes in clusters, Airbase vs MACC. Upper values indicate the mean seasonal amplitude of each cluster. 1$^{st}$ CA.





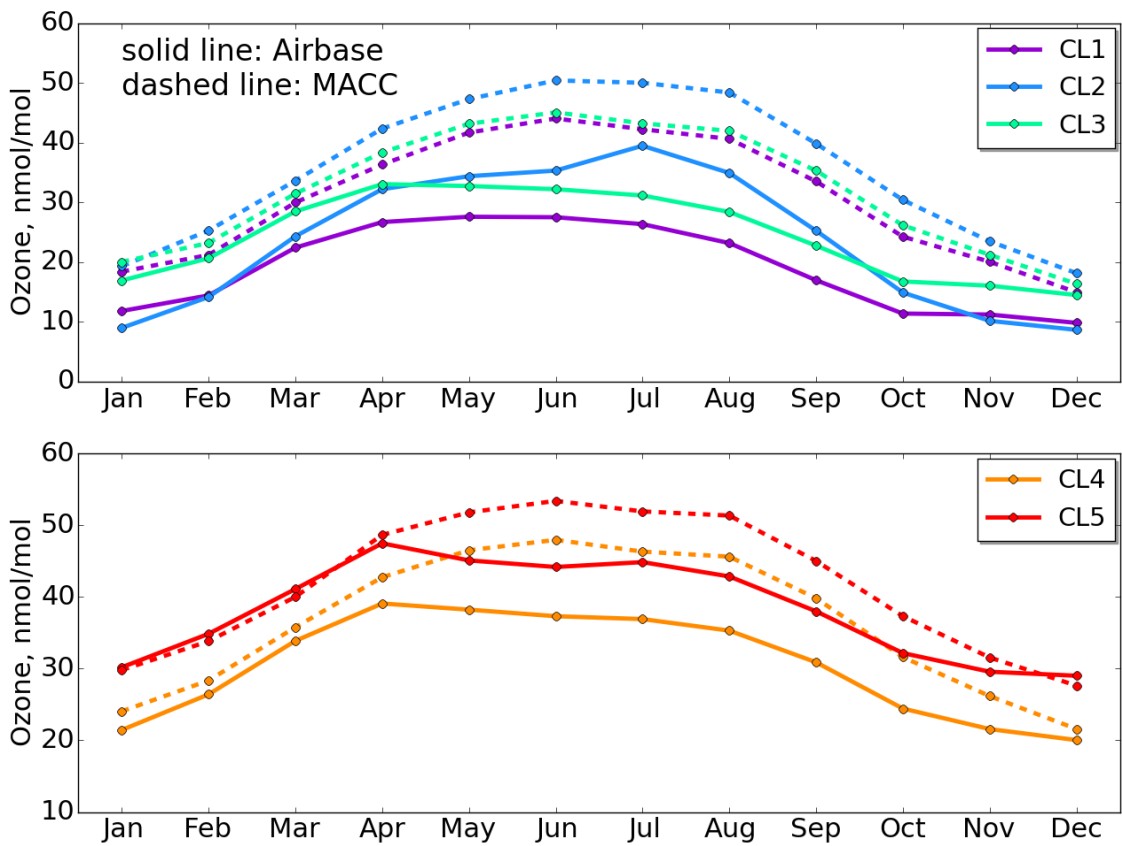

Figure 12. Seasonal cycles of cluster centroids, Airbase vs MACC. 1$^{st}$ CA.



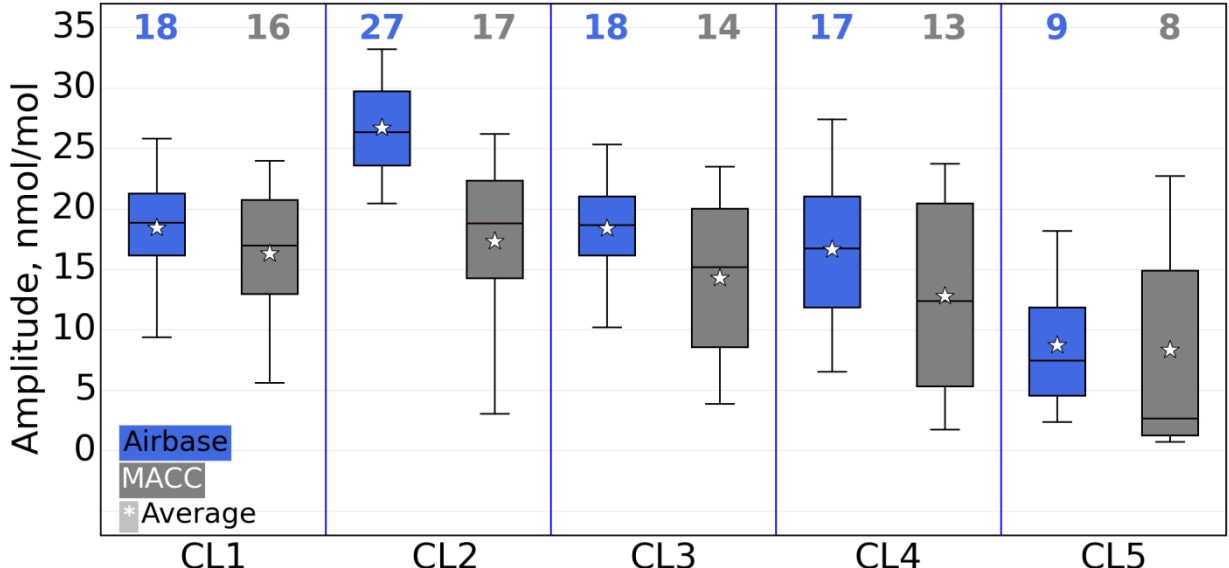

Figure 13. Percentiles (5-25-50-75-95) of ozone diurnal amplitudes in clusters, Airbase vs MACC. Upper values indicate the mean daily amplitude of each cluster. 1[st] CA.





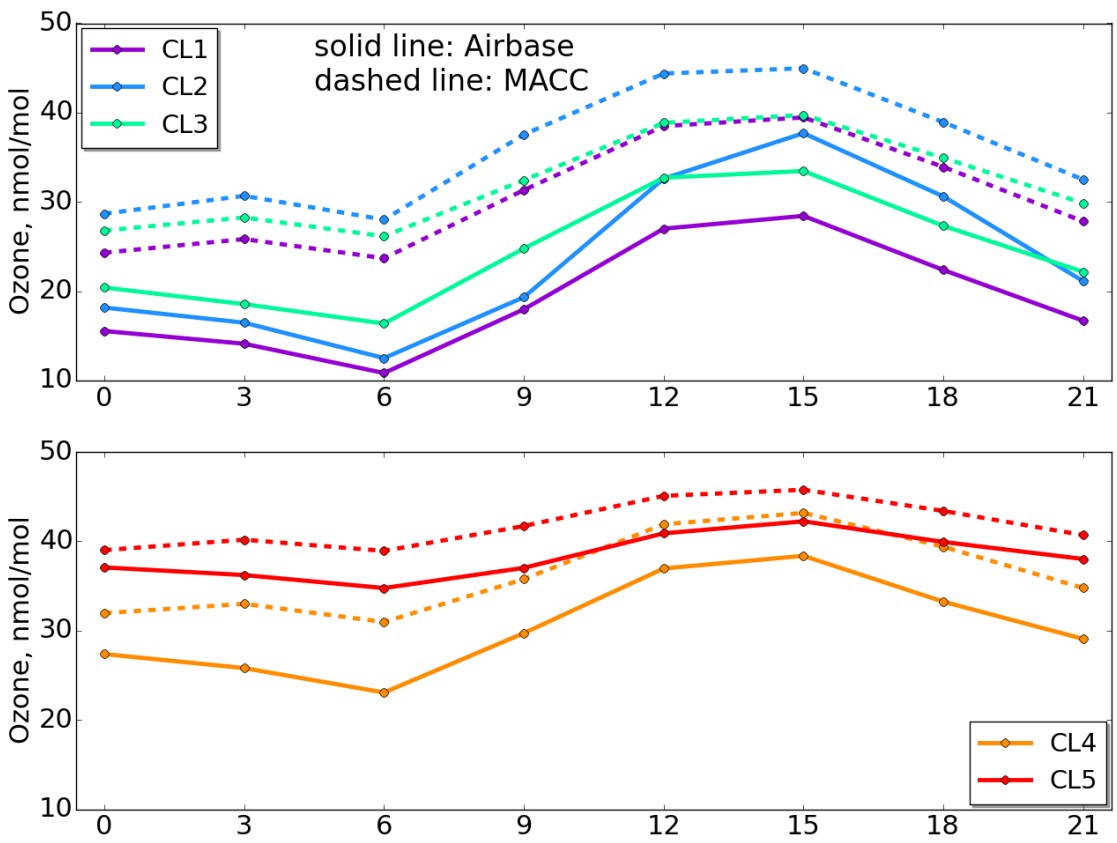

Figure 14. Diurnal cycles of cluster centroids, Airbase vs MACC. 1st CA.





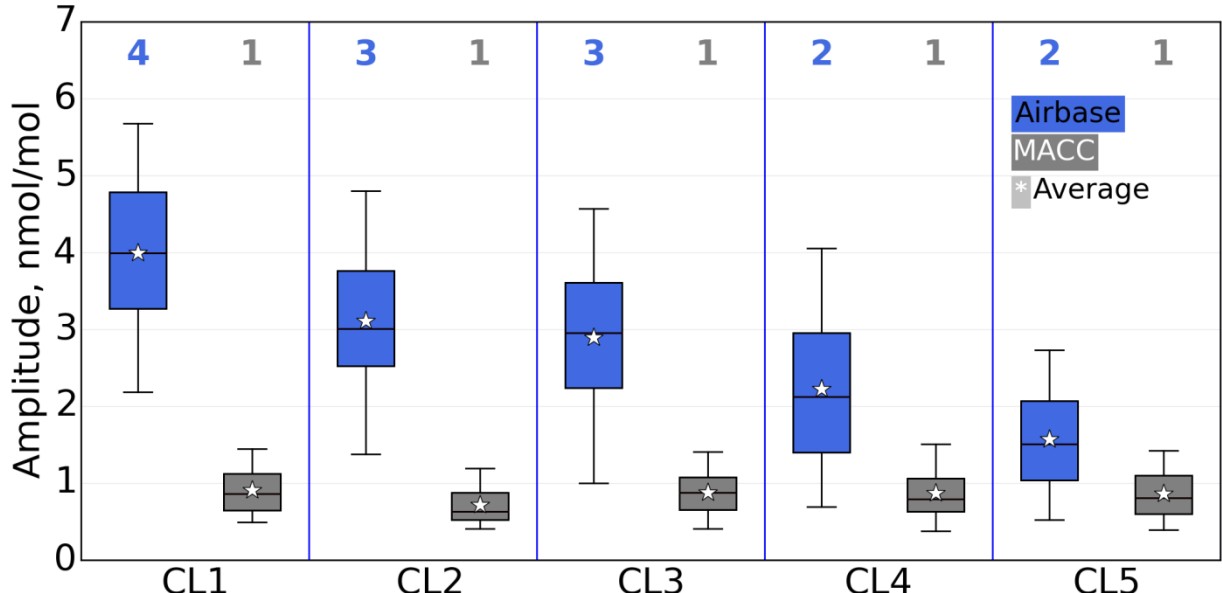

Figure 15. Percentiles (5-25-50-75-95) of ozone weekly amplitudes in clusters, Airbase vs MACC. Upper values indicate the mean weekly amplitude of each cluster. 1st CA.





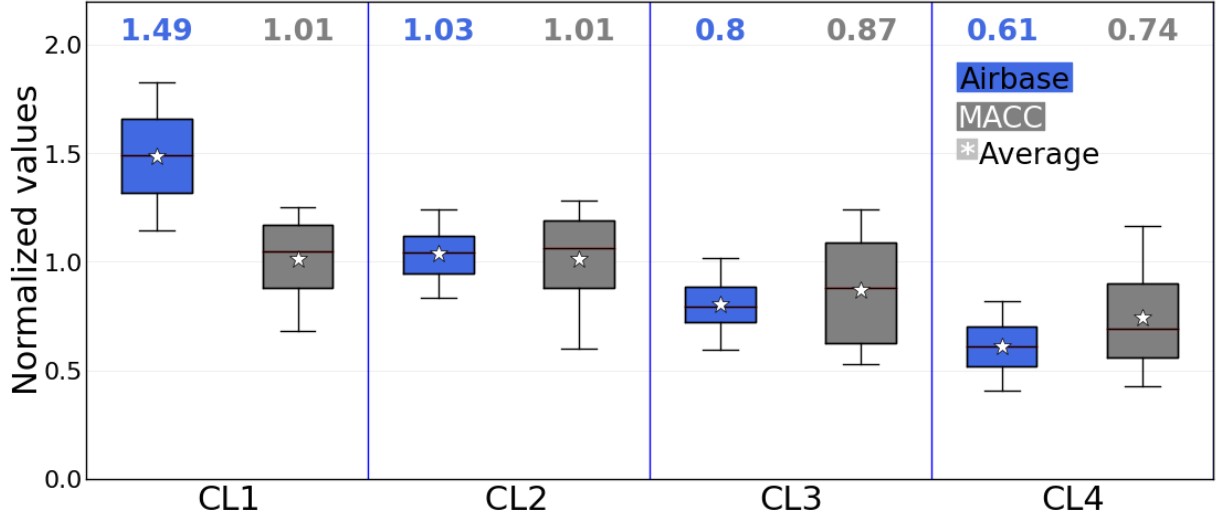

Figure 16. Percentiles (5-25-50-75-95) of ozone seasonal amplitudes in clusters, Airbase vs MACC. Upper values indicate the mean seasonal amplitude of each cluster. 2$^{nd}$ CA.





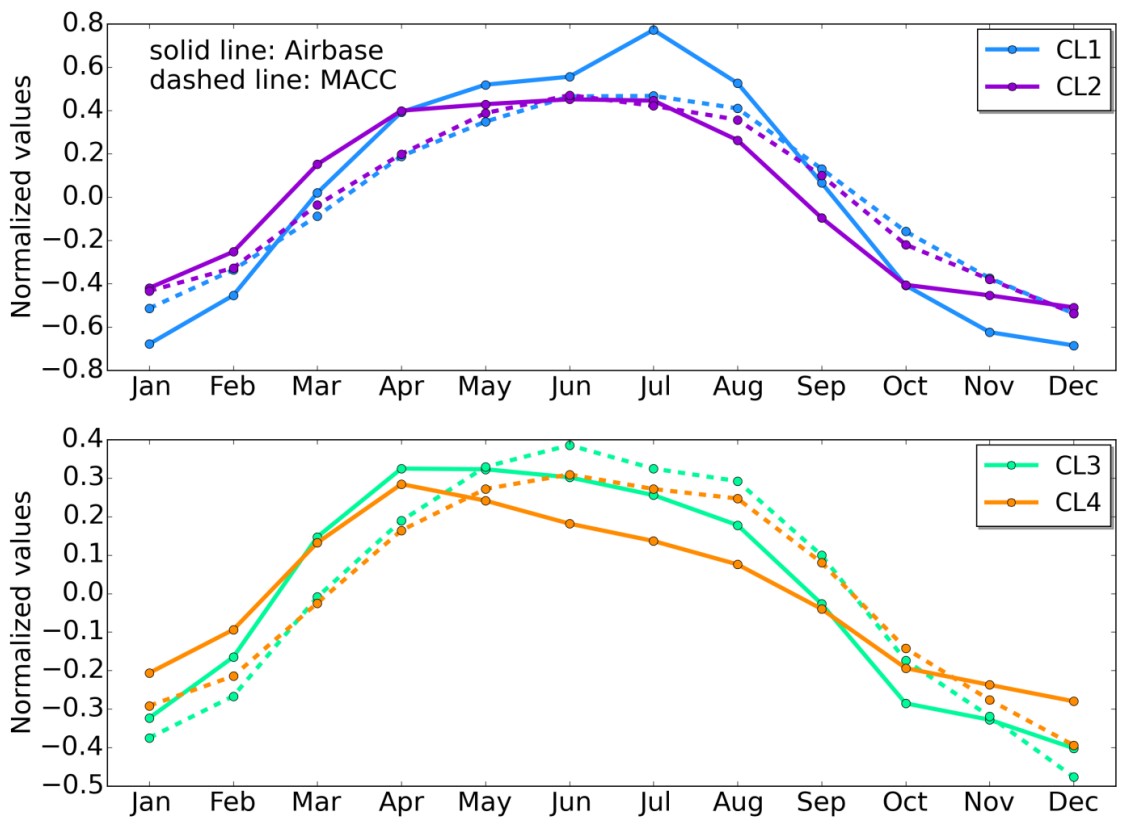

Figure 17. Seasonal cycles of cluster centroids, Airbase vs MACC. 2nd CA.





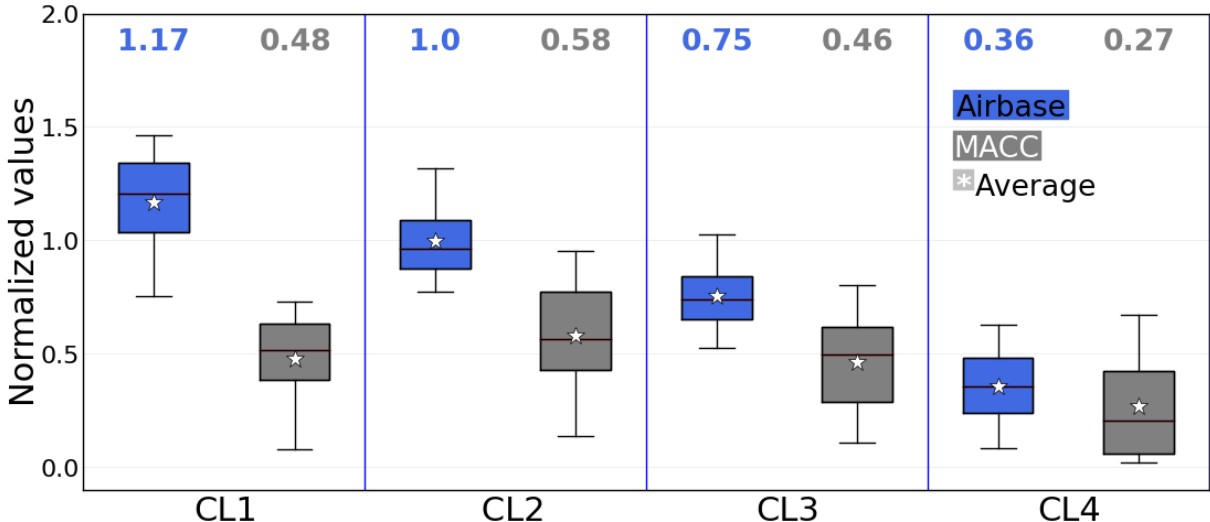

Figure 18. Percentiles (5-25-50-75-95) of ozone diurnal amplitudes in clusters, Airbase vs MACC. Upper values indicate the mean seasonal amplitude of each cluster. $2^{nd}$ CA.

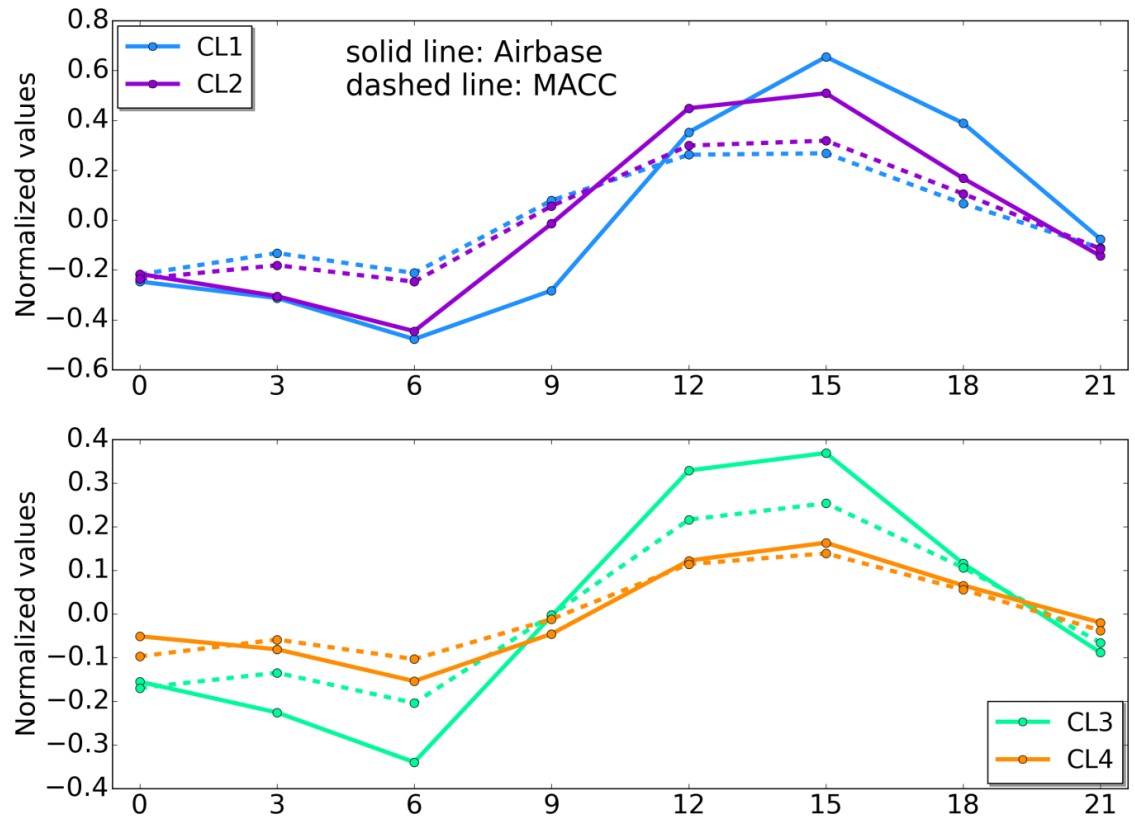

Figure 19. Diurnal cycles of cluster centroids, Airbase vs MACC. 2nd CA.





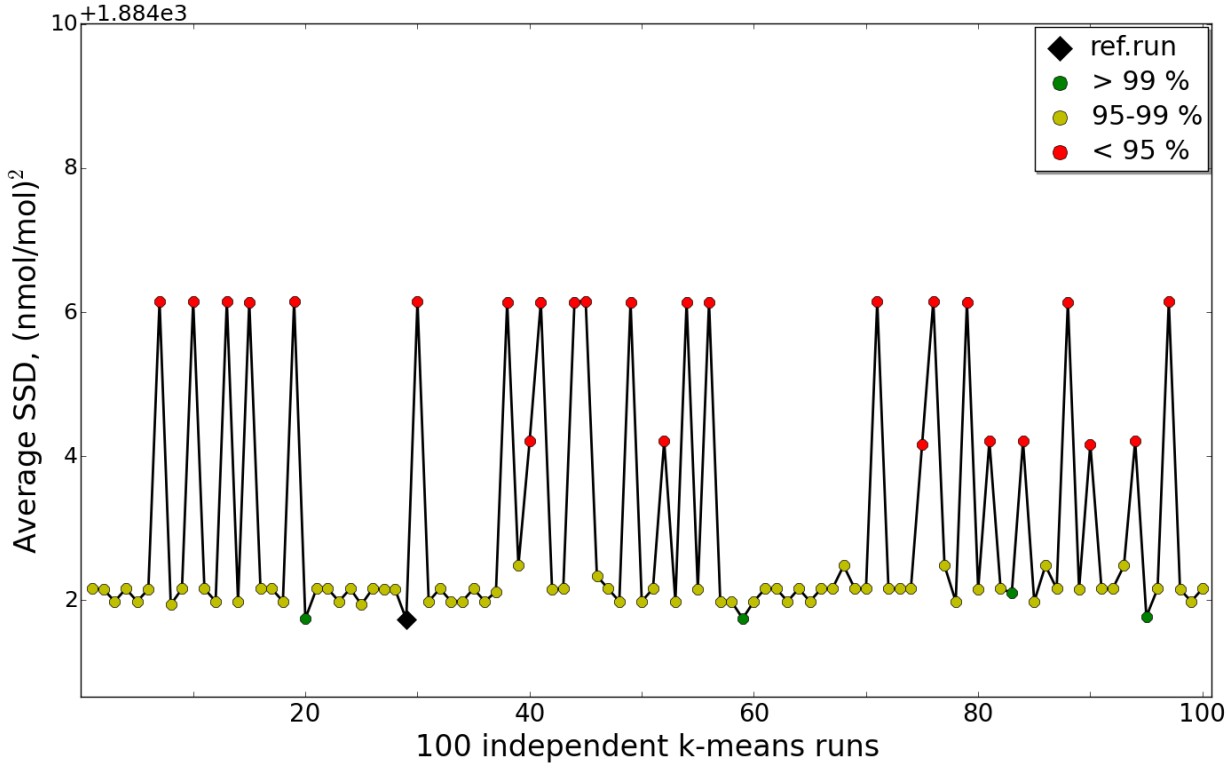

Figure 20. Averaged SSD for 100 independent k-means runs with cluster number k = 5 for all runs. 1st set of properties. Percentage ranges in legend are indicating similarity of corresponding k-means runs with the 1st CA reference run, presented in this work (black diamond dot). First category: 5 runs with > 99% of similarity, second category: 70 runs with 95-99% of similarity and third category: 25 runs with < 95% of similarity (always at least 89% of similarity).





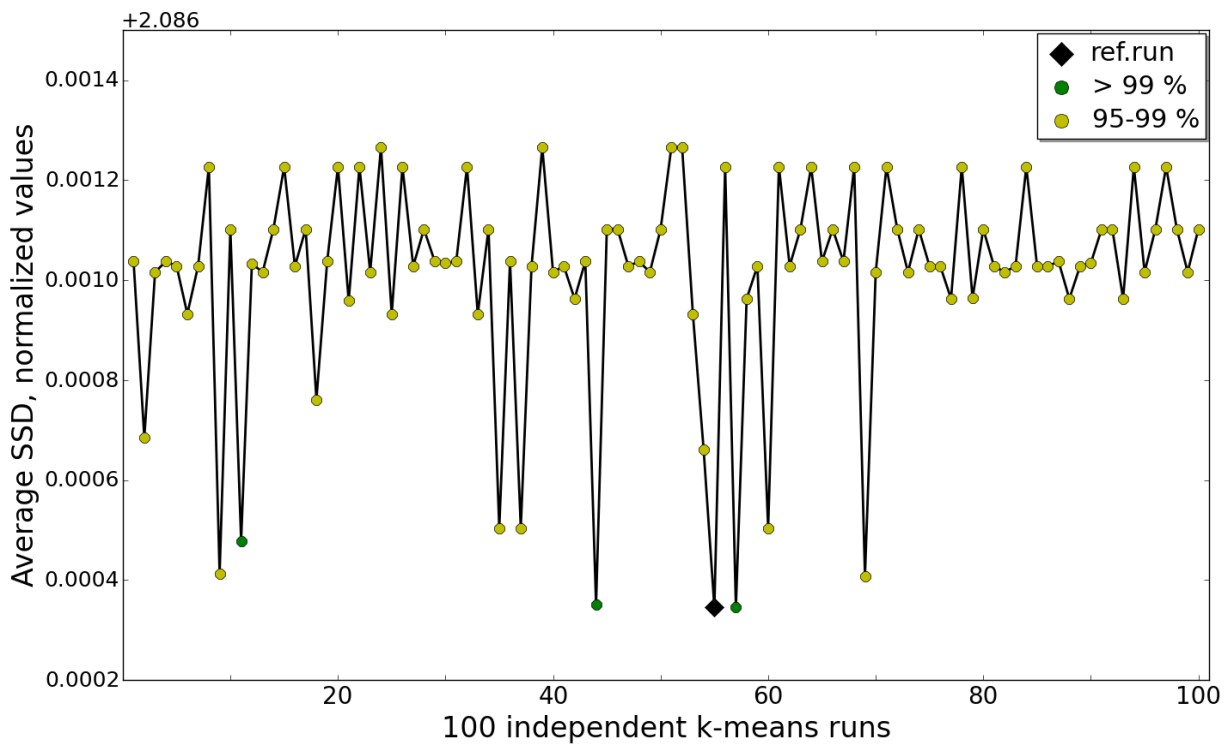

Figure 21. Averaged SSD for 100 independent k-means runs with cluster number k = 4 for all runs. 2$^{nd}$ set of properties. Percentage ranges in legend are indicating similarity of corresponding k-means runs with the 2$^{nd}$ CA reference run, presented in this work (black diamond dot). First category: 4 runs with > 99% of similarity, second category: 96 runs with 95-99% of similarity (always at least 95% of similarity).