# Peer review of "Cluster analysis of European surface ozone observations for evaluation of MACC reanalysis data"

_Atmospheric Chemistry and Physics, 2015_

## Referee Comment (RC1) · Anonymous Referee #1 · 3 Mar 2016

This is a review of the manuscript titled "Cluster analysis of European surface ozone observations for evaluation of MACC reanalysis data" authored by Lyapina et al. The manuscript describes a clustering analysis of MACC model data and Airbase observations for four years using 1492 Airbase sites using three-hourly ozone data over Europe. The authors apply a K-means clustering technique to subset the Airbase sites into five and four clusters, depending on the ozone metric used. The authors compare/contrast the different clusters based on various characteristics of the sites including location, landuse, emissions profile, etc. The work has implications for evaluating models, particularly coarse grid-scale models, since the model data used were at 80km grid spacing, which is much coarser than most regional scale model applications.

General Comments: Overall, the manuscript is very well written and the authors do an excellent job explaining the clusters technique applied and supporting the results of the cluster analysis. In addition, the authors show that the cluster analysis is robust enough to withstand the remove of a large number of the initial sites and be applied to smaller time periods and still retain the integrity of the clusters. It's interesting to see how the different sites across Europe fall into the various clusters. And there does appear to be some regionality to the clusters themselves, although the authors do not focus heavily on that. Overall, I recommend that the manuscript be published with some minor technical corrections and perhaps some clarifications in a few areas.

Specific Comments:

Page 2, Line 1: Provide references for the adverse effects of ozone on human health and agriculture.

Page 8, Line 23: Reword "2nd CA is containing both" to "2nd CA contains both"

Page 8, Line 25: Reword "...cluster is semi-elevated with the mean altitude 433 m for the 2nd CA." to " ...cluster of the 2nd CA with a mean altitude of 433 m."

Page 9, Line 1: Change "...data are both show a positive bias by 9 nmol/mol." to "...data both show a positive bias of 9 nmol/mol."

Page 9, Line 17: Remove the comma after broader.

Page 9, Lines 29-30: Is titration really the only cause of the very low ozone concentrations during the winter? It's seems as though in the winter, which is already non-conducive for ozone formation, that the meteorological conditions could just be very poor for any ozone production to occur. Just suggesting that titration may not be the sole cause of the low ozone concentrations.

Page 10, Line 19: The wording "has twice less probability" is awkward. Suggest re-wording.

Page 10, Line 22: Change "then follow" to "followed by" and "at the end is" to "finally".

Page 10, Line 27: Change "This" to "The".

Page 10, Lines 28-29: Suggesting changing "25%-ile to 75%-ile" to simply "25th to 75th percentiles".

Page 12: It would be useful to the reader to explain exactly how the weekly amplitudes were devised. The analysis seems to suggest that the data were split by day of the week, since the authors refer to specific week days in their discussion. Is that what was done? So more clarification would be helpful here.

Page 12, Lines 28-30: How do the authors know that elevated/residual ozone is the compensating mechanism?

Page 13, Line 20: Change "strong disagreements" to "large differences".

Page 13, Line 22: Change "likewise the distributions" to "likewise for the distributions".

Page 13, Line 23: Change "underestimating observed ones" to "underestimate the observed amplitude".

Page 13, Line 25: Change "give also" to "show".

Page 13, Line 30: Change "gives as a result" to "shows".

Page 13, Line 32: Change "expressed as difference" to "expressed as the difference".

Page 13, Line 33: Change "amplitudes as well as variability decrease" to "amplitudes, as well as variability, decrease".

Page 16, Line 31: Suggest changing "deficits and pros" to "pros and cons".

Page 17, Line 7: Change "mostly always" to "almost always".

Page 17, Line 24: Change "Besides," to "Finally,".

---

## Referee Comment (RC2) · Anonymous Referee #2 · 4 Mar 2016

The manuscript describes a clustering analysis of MACC model data and Airbase observations for four years. A K-means clustering technique was used to group and provide distinctions among various monitors across Europe. The authors provide meaningful explanations for their results and the techniques appear robust. I found the article well written and thought all the main findings were clearly communicated. There was sufficient evidence provided in the data to support the findings. I was especially interested in the explanations for the clustering of certain monitors. the comparison with modeling data was also appropriate. There were some minor editing comments i had, but it appears a previous referee caught nearly all mine and more!

---

## Author Comment (AC1) · 14 Mar 2016

Dear reviewers,

we are grateful for your feedback and valuable comments on our paper. All technical corrections (spelling errors and proposed re-wordings) were taken into account. We would like to reply to the remaining comments as follows:

Referee#1: "Page 9, Lines 29-30: Is titration really the only cause of the very low ozone concentrations during the winter? It's seems as though in the winter, which is already nonconducive for ozone formation, that the meteorological conditions could just be very poor for any ozone production to occur. Just suggesting that titration may not be the

sole cause of the low ozone concentrations."

Thank you for pointing this out. Nevertheless, the statement refers to the extreme low end of the ozone concentration frequency distribution. These almost zero values are primarily explained by titration. To clarify this, we rephrased the sentence as follows: "In the Airbase winter time data the three clusters with more urban characteristics (CL1, CL2 and CL3) contain a significant number of values with very low concentrations, which are primarily caused by ozone titration in the presence of large amounts of NOx from traffic and industries. Peak frequencies are decreasing from CL1 to CL4, though the last is showing only few incidents of "zero" ozone. For clusters CL1, CL3 and CL4 the MACC model is able to capture some of this titration, but not for CL2 (Po Valley). No ozone titration occurs in CL5, neither in the observational data nor in the model results."

Referee#1: "Page 10, Line 19: The wording "has twice less probability" is awkward. Suggest rewording."

"Indeed, the peaks of Airbase probabilities of zero ozone concentrations are pronounced for both clusters in comparison to the moderately polluted CL3, for example, where "zero" ozone occurs only half as often and the ozone maximum appears in the range 25-30 nmol/mol."

Referee#1: "Page 12: It would be useful to the reader to explain exactly how the weekly amplitudes were devised. The analysis seems to suggest that the data were split by day of the week, since the authors refer to specific week days in their discussion. Is that what was done? So more clarification would be helpful here."

We inserted a sentence how weekly amplitudes have been derived: "Weekly amplitudes are shown in Figure 15. These were calculated as the absolute difference between maximum and minimum ozone mixing ratios of averaged weekly cycles for each station and then grouped into clusters accordingly. Weekly amplitudes were not used as initial parameters in the CA, but.... ... The weekly cycles of all cluster centroids

show growth from Friday till Sunday, but no significant change during the week (not shown)." If the reviewers are interested, we present here the resulting figure with the average weekly cycles in all 5 Airbase clusters of the 1st CA (see Figure1 below).

Referee#1: "Page 12, Lines 28-30: How do the authors know that elevated/residual ozone is the compensating mechanism?"

Although we did not analyze this in detail, the more pronounced seasonal amplitudes of CL2 and a separate analysis of summer nighttime and daytime data support our argument. In Figure 2 provided below the normalized frequency distributions of summer nighttime ozone in CL1 show a small peak of very low mixing ratios, characterizing ozone titration, what is not seen for CL2. We rephrased this part as follows: "...CL2 also exhibits ozone titration, but in summer time to a lesser extent than for CL1 (Figure 9). For CL2 ozone destruction by NO and dry deposition still occur during night time but the prevalence of the daily ozone production over the ozone titration is more obvious here than for CL1. Indeed, the seasonal and diurnal cycles of CL2 are more pronounced than for CL1 (Figures 12 and 14), and are indicative of the intensive photochemistry in the Po Valley region. This may be explained by the basin type of the Po Valley and by its partly sub-tropical climate with plenty of available UV light, which is favorable for summer diurnal photochemical ozone production. ..."

Finally, Referee#1 remarks: "...there does appear to be some regionality to the clusters themselves, although the authors do not focus heavily on that."

Indeed, the focus of the paper is placed on the general pollution signatures and robustness analysis of the cluster technique. In the PhD thesis of O. Lyapina (University of Bonn, 2015) an additional analysis investigates whether a more distinct regionalization occurs if the number of k-means clusters is increased to seven, which is still a defendable number based on the SSD tests. While this analysis shows more pronounced regional differences of the resulting clusters, the distinction between them becomes weaker, which makes the interpretation of results with respect to the model evaluation

more difficult.

[Figure]

[Figure]

**Fig. 1.** Weekly cycles of Airbase cluster centroids. 1st CA.

**Fig. 2.** Normalized frequency distributions of 3-hourly ozone values in clusters (2007-2010), summer daytime (left) and summer nighttime (right), Airbase vs MACC. 1st CA.